# Seeing What Matters: Generalizable AI-generated Video Detection with Forensic-Oriented Augmentation

**Riccardo Corvi**[1,2][*], **Davide Cozzolino**[1], **Ekta Prashnani**[2],
**Shalini De Mello**[2], **Koki Nagano**[2], **Luisa Verdoliva**[1]
[1]University Federico II of Naples, [2]NVIDIA

## Abstract

Synthetic video generation is progressing very rapidly. The latest models can produce very realistic high-resolution videos that are virtually indistinguishable from real ones. Although several video forensic detectors have been recently proposed, they often exhibit poor generalization, which limits their applicability in a real-world scenario. Our key insight to overcome this issue is to guide the detector towards *seeing what really matters*. In fact, a well-designed forensic classifier should focus on identifying intrinsic low-level artifacts introduced by a generative architecture rather than relying on high-level semantic flaws that characterize a specific model. In this work, first, we study different generative architectures, searching and identifying discriminative features that are unbiased, robust to impairments, and shared across models. Then, we introduce a novel forensic-oriented data augmentation strategy based on the wavelet decomposition and replace specific frequency-related bands to drive the model to exploit more relevant forensic cues. Our novel training paradigm improves the generalizability of AI-generated video detectors, without the need for complex algorithms and large datasets that include multiple synthetic generators. To evaluate our approach, we train the detector using data from a single generative model and test it against videos produced by a wide range of other models. Despite its simplicity, our method achieves a significant accuracy improvement over state-of-the-art detectors and obtains excellent results even on very recent generative models, such as NOVA and FLUX.

## 1 Introduction

In recent years, the field of AI-based video generation has witnessed rapid advancements. Several flexible tools exist, based on diffusion models, that generate high-quality videos from general conditional inputs like text and images [41, 77, 95]. These powerful tools enable professionals to use AI for innovative and creative applications, such as design, marketing, and entertainment. However, the misuse of such technologies also raises ethical and social concerns related to the spread of disinformation, the violation of intellectual property rights and more generally the erosion of trust in digital media [28, 5, 46]. Therefore, there is an urgent need to develop effective methods for distinguishing real from AI-generated videos.

Previous work in video forensics focused mostly on facial forgery detection [66, 83, 85, 33] and proposed solutions that are specifically tailored to faces, for example, by analyzing head pose or facial landmarks and appearance [54, 15], inconsistencies in skin texture or lip motion [49, 94, 34],

---

[*]Part of the work was done during an internship at NVIDIA.
Project page: https://github.com/grip-unina/WaveRep-SyntheticVideoDetection

39th Conference on Neural Information Processing Systems (NeurIPS 2025).

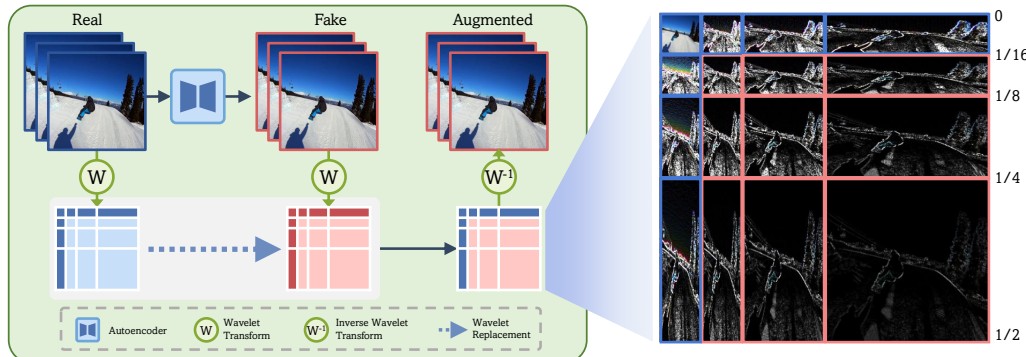

Figure 1: Synthetic video generators leave distinct traces, that are observed in the frequency spectrum. We leverage this observation to enhance their generalizability. To this end, we propose a novel training-time data augmentation strategy based on wavelet-bands that forces the model to learn the frequency components that best distinguish real from synthetic content. Fakes are also generated through video autoencoding to avoid semantic bias and to trick the model into exploiting low-level forensic traces left by the modern video generation architectures. Our training paradigm improves the generalizability of the detector without the need for complex algorithms and large datasets that include multiple generators.

unnatural heart rate variations or motion [17, 24] or identity-based biometrics [1, 61]. However, these approaches are inherently limited to facial content and struggle to generalize to more complex and semantically rich synthetic videos generated by modern diffusion models. Only a few works have been designed so far for fully generated video detection [12, 48, 52, 3, 43]. Such methods focus primarily on designing novel and often complex architectures, neglecting the critical role of the data. In fact, in forensic applications, the choice of training and test data is essential to ensure that detectors learn generation-related cues, rather than spurious correlations encoded in the data [74, 50, 10]. Even though the presence of dataset biases is a well-known risk in machine learning, it still forms a significant and underestimated problem in forensics research. In [9], it was shown that a popular benchmark dataset included real and manipulated images compressed at different JPEG quality levels, causing classifiers to learn compression artifacts instead of tampering cues, leading to poor generalization. This same issue was recently observed for the detection of AI-generated images [31], while other works have pointed out the presence of other possible biases for the same task, such as content or resolution bias [10, 65, 32].

In this work, our aim is to design a synthetic video detector that is truly based on generation-specific artifacts, i.e. traces related to the generation process. This is because methods that rely exclusively on data-driven learning or semantic errors in generation tend to overfit to the training data or to the artifacts introduced by specific generators, e.g., certain semantic visual cues, such as the lack of perspective or temporal consistency in the generated videos, which lead to poor generalization. We overcome this limitation by identifying and integrating priors into the synthetic video detectors that leverage forensic traces that are consistently present across diverse generative models.

To this end, we address the following questions: *i)* What are good discriminative and robust forensic traces present in AI-generated videos? *ii)* What is a good strategy to exploit them? As a first step, we identify the hidden forensic traces that are shared by modern video AI-generators and arise from the generative architectures themselves. Prior art has shown that the up-sampling operations inherent in the synthesis network give rise to quasi-periodic patterns that are clearly visible as peaks in the high-frequency portion of the Fourier spectrum [93, 69]. As a consequence, several works exploited high-frequency traces for AI-generated image detection [30, 47, 10]. Unfortunately, high-frequency components are severely degraded by compression, especially by the strong compression usually adopted for videos, with the effect of washing out the most prominent forensic traces (see Fig.3). However, Diffusion-based generated content differs significantly from natural content not only at high but also at medium frequencies [18]. This latter finding is especially important in this context since we show that video compression impacts less the video's diagonal mid-high frequencies, which therefore turn out to be both discriminative and robust, a good basis to build effective detectors.

To exploit these traces we chose not to work on new detection architectures but, inspired by the recent image-forensics literature [32], to focus on the crucial training phase, in order to emphasize the most

meaningful artifacts and thus guide the detector in learning the right features. To this end, we build a dataset of paired real-fake videos with the same caption and carry out two forms of augmentation (see Figure. 1): 1) we add controlled fake videos by injecting the architecture-related artifacts on real videos through a video autoencoder. This has already been shown to be effective in avoiding possible semantic biases [65, 32]; 2) we push the model to exploit middle frequencies by a tailored augmentation strategy that works on selected bands of a multiscale wavelet decomposition of the video. This has the advantage of avoiding looking at specific compression cues that affect both real and generated videos.

Overall, we make the following contributions:

- we find that inconsistencies in the middle-high (diagonal) frequency content of synthetic video are discriminative, robust, and common across several different video generators, even more recent models;
- we propose a novel augmentation strategy that works on the wavelet bands to guide the model towards exploiting such cues;
- we show that by including this simple strategy we can outperform current SoTA methods in terms of generalization. Across 15 different generators, our approach yields an average improvement of around 12% in terms of accuracy.

## 2 Related Work

**Text-driven video generation.** Synthetic video generation has advanced rapidly in recent years, driven largely by powerful tools like diffusion-based models. Early methods like Text2Video-Zero [41], Modelscope [77], and Hotshot-XL [55] adapted image generators for video synthesis, but struggled with challenges such as temporal coherence and motion consistency. Models—including Mochi-1 [73], Allegro [95], Opensora-Plan [45], and CogVideoX [86]—introduce dedicated architectures leveraging 3D causal autoencoders and 3D transformers, enabling better modeling of spatio-temporal relationships. These models also achieve stronger semantic alignment with input text and produce videos with higher quality and improved temporal consistency [36]. Recently, autoregressive models, such as Loong [81] and NOVA [25], have shown impressive performance for video generation. Although they are based on a completely different generation paradigm, we show that they still exhibit anomalies similar to diffusion models that can be exploited by the forensic classifier.

**Detection of AI-generated videos.** Initial approaches for detecting fully generated videos focused on human motion cues [6], while others adapted image-based forensic detectors using few-shot learning [75]. More recent methods exploit both spatial and temporal artifacts. Bai et al.[3] propose a two-branch CNN to capture spatial anomalies and optical flow inconsistencies, while Chang et al.[11] use three 3D CNNs that target appearance, motion, and geometry. Liu et al. [48] take a different path, feeding RGB frames and reconstruction errors into a CNN+LSTM model, based on the idea that diffusion models can better reconstruct synthetic images than real ones, a principle previously applied to images [82]. Other methods leverage vision-language foundation models like CLIP [52], X-CLIP [12], and LLaVa [71] to capture both spatial and temporal inconsistencies. A more recent approach [43] also introduces a loss function to encourage attention over diverse spatial regions, improving detection beyond facial areas. Differently from such works that focus on the architectural design and train on datasets with standard augmentation, we propose a new training paradigm to capture more robust low-level features, which ensure better generalization across different generative models.

**Frequency artifacts.** AI-generated images often exhibit distinctive Fourier-domain signatures that reveal their synthetic origin. This was shown already for GAN generated images in [93, 69], where the detector was developed to exploit the spectral peaks introduced by the upsampling operations common to such architectures. Building on frequency-domain analysis, prior work [30, 27] trains models on Fourier spectra extracted from real and synthetic images, while [26] shows that GAN-generated images deviate from natural spectral distributions and proposes a simple detector based on the energy spectrum. In [39], a spectral-based adversarial training is proposed to encourage the GAN generator to better reproduce natural spectral distributions, while [35] shows that this training is not sufficient to make GAN-generated images undetectable. Also Diffusion models poorly reconstruct mid-band frequencies compared to real images [18]. This is exploited in [16] that leverages frequency-guided reconstruction to identify the information that the model struggles to reconstruct. Other works force

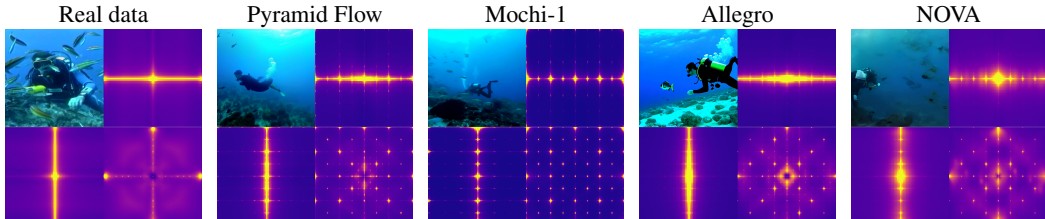

| Real data | Pyramid Flow | Mochi-1 | Allegro | NOVA |

Figure 2: From left to right: a real video from the dataset proposed in [14] and synthetic videos generated using the same associated prompt "A scuba diver in the ocean surrounded by fishes" with Pyramid Flow, Mochi-1, Allegro and NOVA. For each of them we show its spatial power spectrum $S_{yx}(u, v)$ (bottom-right), and its temporal-spatial power spectra $S_{tx}(w, v)$ and $S_{yt}(u, w)$ (top-right and bottom-left).

the network to focus on high-frequencies related features [62, 72] or to exploit relationships of spatial and frequency domains [80]. However, none of these works accounts for the impact of compression in their method design.

**Augmentation strategies.** Data augmentation is essential in computer vision, improving both generalization and robustness by increasing data diversity. It also plays a fundamental role in the detection of synthetic content. Wang et al. [79] showed that augmentation is key to generalizing across unseen generative models, with operations like blurring and JPEG compression being especially effective. Despite its impact, augmentation has received limited attention in the forensic literature. Most existing strategies are borrowed from computer vision and rely on high-level transformations such as brightness and contrast adjustments [7, 19], or cut-out and mix-up techniques [78]. Additional high-level strategies have been proposed for facial manipulations in videos [44, 23]. A different path is pursued in [84] where overfitting to method-specific cues is addressed by proposing a latent space augmentation by simulating variations within and across forgery features. Instead, in [40] it is introduced a Mixup augmentation, which dynamically diversifies frequency characteristics of training samples to mitigate spurious shortcuts and improve generalization. However, we believe that in order to enhance video forensic traces, it is important to design an augmentation strategy that drives the model to focus on the most relevant cues, as it is done for other low-level vision applications [13, 87, 70].

## 3 Proposed Method

### 3.1 Video forensic artifacts

A key step in developing effective detectors for synthetic videos is to identify the most discriminative features. Generators can introduce visual and semantic artifacts, such as geometric distortions, layout inconsistencies, color mismatches, or temporal incoherence. However, these visible errors tend to reduce as generative technologies continue to evolve and it is likely that they may soon disappear altogether. To ensure long-term robustness, our approach focuses on low-level artifacts intrinsic to the generative architecture. In this section we analyze such subtle traces.

**Fingerprints in the Fourier domain.** Early research on synthetic images has shown that, much like real cameras, which imprint each photo with a unique device/camera specific signature [51, 21], synthetic architectures also leave a distinct fingerprint in every generated image [53, 89]. Researchers used these artificial fingerprints to develop methods capable of identifying synthetic images and even trace them back to the specific architecture used for their generation [2]. For synthetic images, these artifacts are clearly visible in the frequency domain [69, 18] and the same holds for frames extracted from synthetic videos [75]. In Fig.2, we show the average power spectra of the residual videos obtained by removing the high-level semantic content of the original videos through a denoiser. More specifically, we compute the residual of the $i$-th video as

$$r_i(m, n, p) = x_i(m, n, p) - \mathcal{D}(x_i(m, n, p); \sigma), \tag{1}$$

where $x_i(m, n, p)$ is the $p$-th frame of the video with size $M \times N \times P$, $m$ and $n$ are spatial coordinates and $\mathcal{D}$ denotes the CNN-based denoiser proposed in [92], which is applied to each frame individually

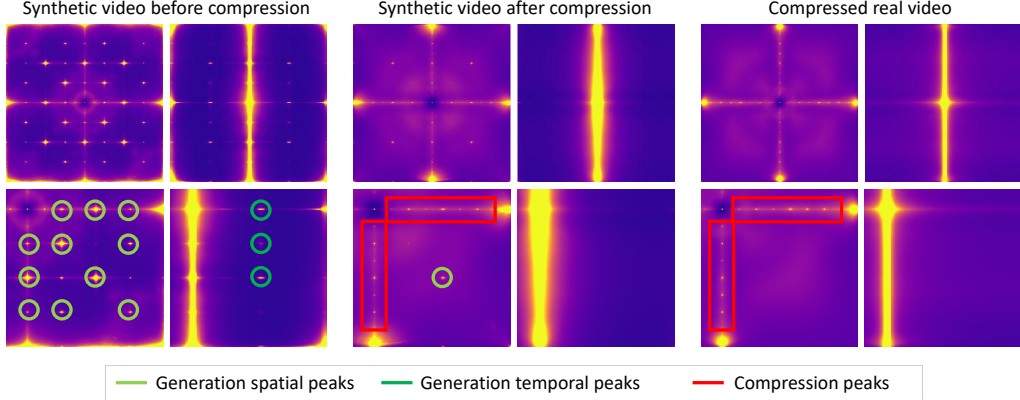

Figure 3: Top: Spatial and temporal-spatial power spectra of OpenSora-Plan videos before and after compression, compared to those computed from a real (compressed) video. Bottom: close up of the power spectra presented on the top. Fourier-domain peaks due to video synthesis (forensic artifacts) are highlighted by circles. Peaks originated by compression are highlighted by red boxes. We can notice that after compression most of the peaks are reduced and compression traces (peaks concentrated along the horizontal and vertical directions) are visible in synthetic videos similar to real ones.

with the noise parameter $\sigma$ set to 1. Then, we evaluate the 3D Fourier transform:

$$R_i(u,v,w) = \sum_{m,n,p=1}^{M,N,P} r_i(m,n,p)e^{-j2\pi(\frac{u}{M}m+\frac{v}{N}n+\frac{w}{P}p)} \tag{2}$$

and compute the spatial power spectrum by averaging the spectra of all frames of $I = 150$ videos:

$$S_{yx}(u,v) = \frac{1}{I}\sum_{i=1}^{I}\frac{1}{P}\sum_{w=1}^{P}|R_i(u,v,w)|^2. \tag{3}$$

The spatial power spectrum accounts for the fraction of the total image power concentrated at a given (vertical, horizontal) frequency pair $(\frac{u}{M}, \frac{v}{N})$. On the other hand, by averaging the power spectra of all videos along the rows we obtain the temporal-spatial power spectrum,

$$S_{tx}(w,v) = \frac{1}{I}\sum_{i=1}^{I}\frac{1}{M}\sum_{u=1}^{M}|R_i(u,v,w)|^2, \tag{4}$$

which accounts for the fraction of the total image power concentrated at a given (temporal, horizontal) frequency pair $(\frac{w}{P}, \frac{v}{N})$. Similarly, we can compute the temporal-spatial power spectrum $S_{yt}(w,v)$ concentrated at a given (vertical, temporal) frequency pair. The spatial power spectra of Fig.2 exhibit typical spectral peaks (visible as bright spots) caused by the upsampling process in the generative architecture. Interestingly, similar peaks are also visible along the temporal direction. Notice, however, that these peaks are not present in the real video (Fig.2, first column).

**Impact of video compression.** In realistic scenarios requiring efficient storage and transmission, synthetically generated videos are usually compressed using standard codecs. These codecs exploit redundancies along both spatial and temporal dimensions, but to achieve higher compression, they also tend to smooth the original signal, thus removing most of the high-frequency components, and valuable forensic artifacts along with them. Indeed, seeing Fig.3, we can note that, after compression, most peaks vanish from the spatial spectra (left) and all of them from the temporal spectra (right). In addition, we can observe that compression, due to its block-wise processing, also introduces some strong peaks of its own, both in synthetic and real videos. This suggests that looking at vertical and horizontal directions could easily trick a detector into wrong decisions. Instead we can observe that the peak along the diagonal direction is still present, and can provide a more meaningful and robust cue. Notably all synthetic generators exhibit such diagonal peaks, even autoregressive models (Fig.2), which survive even after compression (see appendix). This observation motivates our focus on these artifacts, that are both shared across different generators and are robust to compression.

## 3.2 Training strategy

The analysis described above has highlighted the need to focus on artifacts in the mid-high frequency range along the diagonal directions. To this end, it is worth noting that for Diffusion-based generated images diagonal frequencies were found to be more difficult to synthesize than vertical and horizontal ones in [18], which further supports our conjecture. To implement our training strategy, we include two forms of augmentation: the addition of simulated fake videos in the training dataset by fingerprints injection and the inclusion of an augmentation strategy during training based on the wavelet transform.

**Artificial injection of forensic cues.** To encourage the detector to look at low-level artifacts, we add fake videos in the training dataset that are visually indistinguishable from real ones, but embed the traces related to the generation architecture. To achieve this, real videos are reconstructed with the same autoencoder used during generation. This idea is not new, and is actually gaining ground lately. Several works have simulated forensic artifacts to train the detectors both for GAN-based images [93, 37] and for diffusion models [65, 32]. Building fakes in this way also removes the semantic content bias and encourages the detector to identify intrinsic artifacts and patterns related to the generation process versus relying on differences in the semantic content. We adopt the Variational Autoencoder (VAE) used by Pyramid Flow [38]. This is a 3D convolutional neural network trained on the WebVid-10M dataset [4] with a latent space that compresses videos both spatially and temporally using a downsampling ratio of $8 \times 8 \times 8$. Its structure is similar to MAGVIT-v2 [88], incorporating causal convolution in time and pixel shuffle operations for decoder upsampling.

**Augmentation by wavelet-band replacement.** We propose a new form of augmentation that is based on the wavelet decomposition. During training, for fake videos we replace the low-frequency bands (plus those along the horizontal and vertical directions) with those of the real counterpart. In this way, the detector is forced to base the decision on the mid-high frequency diagonal components. The use of wavelet decomposition helps to easily conduct such replacement, since this transform splits the signal into multiple frequency bands preserving spatial relationships within each band (see Fig.1). The augmentation process is summarized in Fig.4 and described in the following. First, we compute the Wavelet Transform. Then the baseband, the vertical and horizontal frequency bands of the fake frame are substituted with those of the

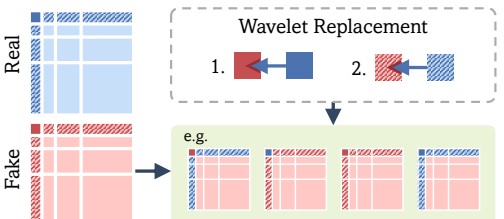

Figure 4: For each real video, four corresponding fake versions are generated: one with all fake subbands and others with selective replacements, such as the baseband or specific subbands substituted with their real counterparts. Notably, diagonal mid-to-high frequency subbands are never replaced: this is to teach the detector to do without the traces brought by one or the other low-frequency subbands and hence focus on such traces.

real counterpart. Finally, the inverse Wavelet Transform is applied. We adopt the Fully Separable Wavelet Transform (FSWT), which offers an anisotropic representation that more effectively localizes frequencies relative to horizontal and vertical discontinuities compared to standard wavelets [76, 59], which is a property particularly useful to our task. We leverage the Haar wavelet transform [60] with three decomposition levels, resulting in the image being split into 16 frequency bands. The two-dimensional FSWT can be implemented by applying the multilevel decomposition separately along the rows and columns of each frame of the video. At each decomposition level $i$, the wavelet transform recursively splits the lowest-frequency band along each dimension into two sub-bands. This is achieved using two convolutions with a stride of 2:

$$x_i^L = x_{i-1}^L \otimes_s h_L \qquad x_i^H = x_{i-1}^L \otimes_s h_H,\qquad(5)$$

where $\otimes_s$ denotes the stride convolution operation, $h_L$ and $h_H$ are the kernels of two convolutions. At each level $i$, the low frequency band, $x_i^L$, represents a low-resolution approximation of the input initialized with the original data at the first iteration. The high frequency band, $x_i^H$, captures detail information initially present in the higher resolution data. In a similar way, the inverse wavelet decomposition is based on transposed convolutions:

$$x_{i-1}^L = x_i^L \otimes^T h_L + x_i^H \otimes^T h_H,\qquad(6)$$

where $\otimes^T$ denotes the transposed convolution operation with a stride of 2. Wavelet decomposition also offers computational advantages, as its structure is well-suited for efficient implementation on

GPUs and also has a complexity of $O(n)$ less than the $O(n \cdot \log(n))$ of the Fourier transform. The two kernels for the Haar wavelet are: $h_L = \frac{1}{\sqrt{2}}[1 \ 1] \quad h_H = \frac{1}{\sqrt{2}}[-1 \ 1]$.

## 4 Results

### 4.1 Experimental setup

**Architecture.** As already discussed previously, we do not propose a new architecture and leverage the power of large pretrained models, that have already shown their potential for the detection of synthetic media [56, 20, 52, 71, 12]. In particular, we consider a DINOv2 model with registers [57, 22] trained end-to-end and frame-by-frame. The final decision is made by averaging the logit scores of 64 frames. Additional details about the model we used and our implementation are described in Section D.

**Training data.** To test for the generalization ability of our method, we train the model using one single synthetic generator. This is a common practice in forensics, reflecting the realistic scenario where generative architectures are unknown at test time. We leverage the Pyramid Flow model since it can generate synthetic videos with no clear visible errors and it was possible to regenerate artificial synthetic data through reconstruction (see Section 3.2). Real videos come from the Panda70M dataset [14]. Overall, training / validation datasets include 4,200 / 900 videos. Pairs of real and generated videos are carefully constructed using identical prompts to reduce content bias. To avoid format bias, all generated videos are compressed using the same codec as the real videos, i.e. H.264, as suggested in [31]. More information on the training data are included in the appendix.

**Test data.** We benchmark our model on the publicly available GenVideo dataset [12] which, includes videos generated by several models for a total of around 20k real and fake videos. We decided not to use the Diffusion Video Forensics (DVF) dataset [71], since real videos are compressed using MPEG-4 Part2, while fake videos are mostly compressed using H.264. This difference in format introduces a bias that can be exploited if the detector is not properly trained, leading to an incorrect performance evaluation [31]. To test on more recent models we created a dataset of 2,400 videos including the following generators Allegro [95], CogVideo X1.5 [86], Mochi-1 [73], OpenSora-Plan [45], Sora [8], NOVA [25] and FLUX [29] (more details in the appendix). It is worth noting that NOVA does not model a joint distribution as diffusion models do; instead, it adopts a non-quantized autoregressive formulation. For this reason, it represents an excellent test of a detector's ability to generalize to different generation strategies. Also for the test data, real and generated video pairs share identical prompts and all synthetic videos are compressed with H.264 (same codec as for real videos) to avoid any format bias.

**Metrics.** Following previous works, we evaluate performance using the Area Under the Curve (AUC), which does not need to set a decision threshold, and balanced accuracy, which accounts for both false alarms and missed detections. For balanced accuracy, the decision threshold is set at 0.5. We also report the probability of detection at a 5% false alarm rate (Pd@5%), the balanced Negative Log-Likelihood (NLL) [63], and binary Expected Calibration Error (ECE) [58]. The former provides insight into the false alarm behavior, which is critical in forensic applications. The latter two measures are calibration metrics and indicate how much the model's predicted probability scores differ from the true probabilities and quantifies the generalization ability of the model.

### 4.2 Comparison with the State-of-The-Art

We compare our proposed method against several SoTA approaches. Among these, DMID [19], UnivFD [56], RINE [42], FreqNet [72], and FIRE [16] were originally developed for the detection of synthetic images. Others, AIGVDet [3], DeMamba [12], and MM-Det [71], were specifically designed for detecting AI-generated videos. To ensure a fair comparison, we include only methods with publicly available code and/or pre-trained models. All models are trained on the datasets used in their respective original publications: ProGAN for UnivFD and FreqNet, Latent Diffusion for DMID and RINE, ADM for FIRE, Stable Video Diffusion for MM-Det, Moonvalley for AIGVDet, and videos from ten generators (ZeroScope, I2VGen-XL, Stable Diffusion, Stable Video Diffusion, VideoCrafter, Pika, DynamiCrafter, SEINE, Latte, OpenSora) for DeMamba.

**Analysis with methods trained on their original datasets.** Balanced accuracy results across all tested generators are reported in Table 1, while AUC and Pd@5 scores are presented more concisely in Fig.5. We begin our analysis with the average results, shown in the last column of Tab.1. The

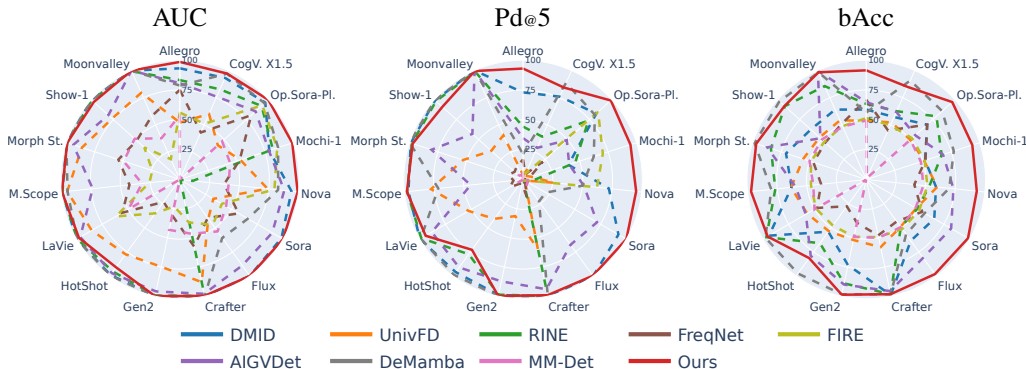

Figure 5: Comparison with SoTA methods on 16 generative models across different evaluation metrics.

proposed method achieves an average balanced accuracy of 94.3%, outperforming DeMamba, the second-best method, by ten percentage points. As expected, methods originally developed for image detection are not competitive in this setting, like UnivFD and FreqNet. However, also the video-oriented AIGVDet lags nearly 20 percentage points behind our proposed method. The performance of MM-Det, falling below random guessing, may seem especially surprising. However, this outcome is primarily due to the biased DVF (Diffusion Video Forensics) dataset used for its original training, which leads to a misleading assessment. Turning to results on the individual generators, a clear divide can be observed between the older ones, included in the GenVideo dataset, and the more recent ones added here. Several detectors exhibit a very good performance on GenVideo (DeMamba is the top detector on those data), which drops substantially on the newer generators, highlighting a limited generalization capability. In contrast, the proposed method performs consistently well across all generators, with isolated dips on CogVideo X1.5 and on Hot Shot.

**Analysis with methods re-trained on Pyramid Flow.** To demonstrate that the observed performance gap is not due to the inclusion of the recent Pyramid Flow model in training, we retrained all methods using videos generated by Pyramid Flow. The resulting accuracy metrics are reported in Table 2 (other evaluation metrics are included in the appendix). Under this new training condition, we observe significant changes in performance; however, even with re-training, our method is still better than SoTA methods by a large margin. Results become generally more consistent across both older and newer generators, and in some cases, most notably for MM-Det, substantially better than before. The most striking outcome, however, is that image-oriented methods outperform video-oriented ones and by a notable margin. This is an unexpected result and highlights that the video-based detectors are not able to fully exploit the advantage of operating in 3D. We can observe that other frequency-based approaches, such as FreqNet and FIRE, struggle to achieve good results even if trained on our same dataset. This highlights the importance to properly take into account the influence of the video codec in the proposed strategy. Finally, we also note that a few abrupt drops in performance (e.g., AIGVDet falling from 99.6% to 50.0% on Moon Valley, and DeMamba from 97.4% to 62.0% on Hot Shot) have a significant impact on the averages. These anomalies may be attributed to biases introduced during training, however this also requires further investigation.

Table 1: Comparison in terms of balanced accuracy with SoTA methods trained on their original datasets.

| Method | Recent Generators (2024-25 years) | | | | | | | GenVideo (2022-23 years) | | | | | | | | AVG |
| | Allegro | CogV. X1.5 | OSora Plan | Mochi1 | Nova | Sora | Flux | Crafter | Gen2 | Hot Shot | LaVie | Model Scope | Morph Stu. | Show1 | Moon valley | |
|---|---|---|---|---|---|---|---|---|---|---|---|---|---|---|---|---|
| DMID | 58.7 | 57.3 | 68.7 | 51.8 | 58.8 | 65.8 | 63.5 | **99.0** | 73.4 | 54.1 | 94.2 | 62.9 | 70.7 | 62.6 | 64.1 | 67.0 |
| UnivFD | 49.3 | 52.3 | 49.7 | 51.0 | 59.8 | 48.3 | 49.0 | 58.0 | 51.9 | 53.5 | 57.2 | 63.3 | 55.6 | 52.7 | 54.2 | 53.7 |
| RINE | 63.3 | 62.7 | 79.2 | 66.5 | 53.2 | 49.3 | 49.3 | 98.3 | 89.1 | 66.0 | **96.7** | 76.6 | 84.0 | 91.8 | 85.7 | 74.1 |
| FreqNet | 66.8 | 45.0 | 68.8 | 52.2 | 40.5 | 54.2 | 43.5 | 47.9 | 37.2 | 27.7 | 47.7 | 38.0 | 46.5 | 46.2 | 50.9 | 47.5 |
| FIRE | 51.5 | 49.8 | 51.7 | 51.5 | 51.5 | 51.3 | 51.3 | 49.4 | 49.1 | 47.6 | 48.6 | 46.6 | 48.8 | 47.3 | 48.9 | 49.7 |
| AIGVDet | 64.8 | 62.0 | 55.3 | 67.7 | 73.5 | 82.2 | 81.2 | 95.6 | 90.0 | 79.1 | 58.5 | 59.5 | 87.2 | 51.7 | **99.6** | 73.9 |
| DeMamba | 56.2 | **90.8** | 85.7 | 77.2 | 70.0 | 56.5 | 56.8 | 98.4 | **98.3** | **97.4** | **97.6** | 80.1 | **98.4** | **97.9** | 98.3 | 84.0 |
| MM-Det | 49.5 | 1.3 | 49.8 | 48.2 | 48.2 | 47.8 | 49.3 | 49.9 | 49.3 | 0.1 | 50.1 | 44.6 | 50.0 | 50.1 | 48.4 | 42.4 |
| Ours | **91.5** | 85.0 | **97.0** | **93.5** | **93.2** | **98.5** | **98.0** | 98.3 | **98.8** | 81.4 | 95.5 | **97.1** | 97.0 | 92.1 | 98.4 | **94.3** |

Table 2: Comparison in terms of balanced accuracy with SoTA methods re-trained using Pyramid Flow (*).

| Method | Recent Generators (2024-25 years) | | | | | | | GenVideo (2022-23 years) | | | | | | | | AVG |
|---|---|---|---|---|---|---|---|---|---|---|---|---|---|---|---|---|
| | Allegro | CogV. X1.5 | OSora Plan | Mochi1 | Nova | Sora | Flux | Crafter | Gen2 | Hot Shot | LaVie | Model Scope | Morph Stu. | Show1 | Moon valley | |
| DMID* | 77.3 | 72.0 | 91.2 | 75.8 | 82.0 | 89.2 | 89.7 | 93.7 | 94.6 | 75.1 | 88.3 | 76.9 | 81.0 | 73.6 | 97.0 | 83.8 |
| UnivFD* | 76.3 | 63.8 | 74.5 | 71.5 | 75.2 | 88.7 | 80.3 | 90.4 | 92.8 | 66.6 | 84.9 | 80.6 | 85.7 | 79.4 | 95.0 | 80.4 |
| RINE* | **90.7** | 64.3 | 89.8 | 77.3 | 83.3 | 81.8 | 80.8 | 94.5 | 95.2 | 76.6 | 89.6 | 87.2 | 89.3 | 80.4 | 97.2 | 85.2 |
| FreqNet* | 84.3 | 65.5 | 86.3 | 77.5 | 69.0 | 64.5 | 56.5 | 52.1 | 58.5 | 42.4 | 64.7 | 42.8 | 56.5 | 63.0 | 65.0 | 63.3 |
| FIRE* | 75.3 | 65.8 | 63.3 | 72.8 | 74.8 | 64.0 | 73.7 | 83.2 | 88.0 | 73.8 | 75.6 | 75.0 | 81.4 | 76.0 | 93.1 | 75.7 |
| AIGVDet* | 75.7 | 60.8 | 50.8 | 73.3 | 70.7 | 68.5 | 67.5 | 87.7 | 85.5 | 54.0 | 50.4 | 54.1 | 78.5 | 50.0 | 50.0 | 65.2 |
| DeMamba* | 83.3 | 61.3 | 91.8 | 79.2 | 68.7 | 73.0 | 76.0 | 87.5 | 92.2 | 62.0 | 79.6 | 68.6 | 83.8 | 72.6 | 92.4 | 78.1 |
| MM-Det* | 56.8 | 49.8 | 71.5 | 70.5 | 70.5 | 76.7 | 72.8 | 84.5 | 87.2 | 40.5 | 85.8 | 63.3 | 76.5 | 80.8 | 87.0 | 71.6 |
| Ours | **91.5** | **85.0** | **97.0** | **93.5** | **93.2** | **98.5** | **98.0** | **98.3** | **98.8** | **81.4** | **95.5** | **97.1** | **97.0** | **92.1** | **98.4** | **94.3** |

Table 3: Influence of the augmentation.

| +Rec. | augment. | Allegro | CogVideo X1.5 | Mochi-1 | OpenSora-Plan | AVG |
|---|---|---|---|---|---|---|
| | | **AUC ↑ / bAcc ↑** | | | | |
| | - | 94.7 / 68.2 | 91.3 / 60.2 | 94.9 / 75.5 | 98.0 / 80.0 | 94.7 / 71.0 |
| ✓ | - | 97.3 / 87.8 | 95.0 / 78.2 | **98.9 / 90.8** | **99.4 / 95.8** | 97.6 / 88.2 |
| ✓ | MixUp | 96.7 / 84.7 | 94.8 / 79.0 | 98.6 / 87.8 | 99.3 / 94.3 | 97.3 / 86.5 |
| ✓ | CutMix | **97.8 / 87.7** | 95.8 / 80.3 | **98.9 / 92.3** | 99.3 / 94.3 | 97.9 / 88.7 |
| ✓ | WaveRep | **98.6 / 91.5** | **97.2 / 85.0** | 99.1 / 93.5 | **99.8 / 97.0** | **98.7 / 91.8** |
| | | **NLL ↓ / ECE ↓** | | | | |
| | - | 1.95 / .309 | 2.73 / .389 | 1.58 / .244 | 1.03 / .196 | 1.82 / .284 |
| ✓ | - | 0.50 / .118 | 0.88 / .198 | 0.28 / .084 | 0.15 / .038 | 0.45 / .109 |
| ✓ | MixUp | 0.35 / .106 | 0.71 / .197 | 0.24 / .094 | 0.12 / .037 | 0.35 / .109 |
| ✓ | CutMix | 0.38 / .120 | 0.63 / .185 | 0.22 / .073 | 0.15 / .051 | 0.35 / .107 |
| ✓ | WaveRep | **0.28 / .076** | **0.53 / .140** | **0.19 / .056** | **0.07 / .023** | **0.27 / .074** |

Table 4: Influence of the backbone.

| Network | Allegro | CogVideo X1.5 | Mochi-1 | OpenSora-Plan | AVG |
|---|---|---|---|---|---|
| | **AUC ↑ / bAcc ↑** | | | | |
| DINOv2 e2e | **98.6 / 91.5** | **97.2 / 85.0** | **99.1 / 93.5** | **99.8 / 97.0** | **98.7 / 91.8** |
| DINOv2 LP | 86.6 / 79.2 | 77.5 / 70.8 | 88.6 / 81.7 | 91.6 / 83.5 | 86.1 / 78.8 |
| Clip e2e | **99.1 / 86.5** | **97.2 / 67.3** | **98.6 / 79.3** | **99.9 / 92.7** | **98.7 / 81.5** |
| Clip LP | 91.8 / 70.8 | 89.3 / 67.5 | 87.8 / 69.8 | 94.1 / 76.5 | 90.7 / 71.2 |
| Hiera video | **98.9 / 79.7** | 93.5 / 65.8 | 94.7 / 65.3 | **99.4 / 81.8** | 96.6 / 73.2 |
| | **NLL ↓ / ECE ↓** | | | | |
| DINOv2 e2e | **0.28 / .076** | **0.53 / .140** | **0.19 / .056** | **0.07 / .023** | **0.27 / .074** |
| DINOv2 LP | 0.50 / .110 | 0.58 / **.059** | 0.46 / .109 | 0.43 / .126 | 0.49 / .101 |
| Clip e2e | 0.52 / .141 | 1.54 / .312 | 0.92 / .209 | 0.21 / .073 | 0.80 / .184 |
| Clip LP | 0.54 / .207 | 0.60 / .227 | 0.60 / .215 | 0.45 / .173 | 0.55 / .205 |
| Hiera video | 0.66 / .204 | 1.66 / .339 | 1.54 / .338 | 0.49 / .179 | 1.09 / .265 |

## 4.3 Ablation study

Here we present our ablation study, where we analyze our method with different augmentations, backbones and training data.

**Influence of augmentation.** Table 3 shows the impact of different augmentation strategies, using always the same backbone, DINOv2 [57, 22] trained in an end-to-end manner. In the first row, only a standard form of augmentation is used in training, including compression, blurring and resizing, operations typically carried out in forensics applications. Then, starting with the second row, reconstructed fake videos paired with real ones are included (marked by a check in the +*Rec.* column), and new forms of augmentation are added: MixUp [91], CutMix [90], and finally our wavelet-band replacement (WaveRep). On average, across all generators, it appears that the inclusion of artificial fake videos in training (second row) ensures a significant gain in performance under all metrics, AUC and bAcc in the top part of the table, and NLL and ECE in the bottom. Then, the computer vision-oriented forms of augmentation, MixUp and CutMix, provide very limited additional improvements, if any. On the contrary, the proposed WaveRep provides a further performance boost, driving AUC to almost 99% and bAcc past 90%.

**Influence of the backbone.** Next, we fix the training setup, including artificial fake videos and adding the WaveRep augmentation on top of the basic configuration, and analyze performance across different backbone architectures, with results reported in Table 4. The best results for each column (considering a margin of 1%) are highlighted in bold. Focusing on the average performance metrics, we observe that the end-to-end training (e2e) always performs significantly better than linear probing (LP) versions of both DINOv2 and Clip [64]. In the former the entire network is fine-tuned, while only a linear layer on the final features is learned for the latter. We hypothesize that the semantics-oriented pretraining of these models is not well suited to our forensic application, highlighting the need for a deeper adaptation. Excluding the LP variants, DINOv2 stands out as the clearly preferable option across all metrics. In general, the variability in AUC across the variants is more limited, likely because scores are already very close to 100%, but DINOv2 maintains a consistent advantage also compared to a spatial-temporal backbone based on Hiera [67], that works on clips of 16 frames, and the final decision is obtained by averaging the logits across all clips.

Table 5: Performance by varying the number of generators during training in terms of AUC and balanced Accuracy. For each row, the results for the generators used in training are crossed-out. The average values do not include crossed-out results. We compare DINOv2 network with and without our approach.

| # | aug. | Pyramid Flow | CogV. X1.5 | Allegro | OpenSora-Plan | Mochi-1 | Nova | Sora | Flux | AVG |
|---|------|--------------|------------|---------|---------------|---------|------|------|------|-----|
| | | **Recent Generators (2024-25 years)** | | | | | | | | |
| 1 | | ~~100. / 99.7~~ | 91.3 / 60.2 | 94.7 / 68.2 | 98.0 / 80.0 | 94.9 / 75.5 | 99.0 / 87.7 | 95.7 / 69.5 | 96.8 / 72.0 | 95.8 / 73.3 |
| 1 | ✓ | ~~100. / 99.3~~ | **97.2 / 85.0** | **98.6** / 91.5 | **99.8 / 97.0** | **99.1 / 93.5** | **99.3** / 93.2 | **99.9 / 98.5** | **99.9 / 98.0** | **99.1 / 93.8** |
| 2 | | ~~99.9 / 98.5~~ | ~~99.8 / 98.3~~ | 94.9 / 80.5 | 98.7 / 94.7 | 96.5 / 86.5 | **99.4** / 96.5 | 94.4 / 77.5 | 96.8 / 88.2 | 96.8 / 87.3 |
| 2 | ✓ | ~~100. / 99.3~~ | ~~100. / 99.2~~ | **99.1 / 94.3** | **99.8 / 96.5** | **99.6 / 93.5** | **99.9 / 98.7** | **99.9 / 97.8** | **99.9 / 98.8** | **99.7 / 96.6** |
| 4 | | ~~100. / 98.3~~ | ~~99.8 / 97.7~~ | ~~99.9 / 98.0~~ | ~~100. / 98.3~~ | 98.0 / 89.5 | **99.9 / 98.2** | 98.1 / 89.8 | **99.4** / 95.2 | 98.8 / 93.2 |
| 4 | ✓ | ~~100. / 99.2~~ | ~~100. / 98.7~~ | ~~100. / 99.2~~ | ~~100. / 99.2~~ | **99.8 / 97.3** | **100. / 99.0** | **100. / 99.0** | **100. / 99.2** | **99.9 / 98.6** |

**Influence of number of generators in training.** In this section, we evaluate the influence of the number of generators included during training. More specifically, beyond Pyramid Flow, we also add CogVideo X1.5 (two generators) and another setting with four models by adding Allegro and OpenSora Plan. Note that for all these models we considered fake videos reconstructed with the same autoencoder used during generation. Results of our method with and without the wavelet-based augmentation on recent generative models are presented in Table 5, where we crossed-out the results for the generators included during training. We can observe a consistent improvement with the increase of the number of generators. Our proposed augmentation also guarantees a significant improvement especially when few models are considered, though even with four generators there is an increase of around 5% in terms of balanced accuracy. Finally, we want to highlight that our approach ranked first on the SAFE Synthetic Video Detection Challenge 2025 [68]. This challenge simulates a realistic scenario where no information about the test models are known in advance and videos vary in resolution, bitrate, frame rate, and length. This confirms the advantage of our training data and augmentation strategy to ensure good generalization ability.

# 5   Discussion

**Conclusions.** In this paper, we propose a novel augmentation strategy for the detection of AI-generated videos. We begin by identifying key forensic cues that are robust to compression artifacts and consistently present across different generative models. Our approach is motivated by the observation that vertical and horizontal frequency components are particularly susceptible to degradation due to block-based compression algorithms commonly employed in video codecs. Through extensive experiments, we demonstrate that this simple augmentation improves detection performance even if only one single generator is included during training. Our findings highlight the importance of working on the training paradigm and of injecting carefully curated domain specific knowledge, which can yield a greater benefit than the design of more complex detection architectures. We hope this work encourages further exploration of principled training strategies and the discovery of universally present discriminative forensic traces within the forensic community.

**Limitations.** The method proposed in this work incorporates a forensic-oriented augmentation strategy that encourages the detector to focus on the most relevant spatial traces. However, it is not explicitly designed to capture temporal artifacts. Even if our approach outperforms existing methods that leverage temporal information, we believe it is essential to systematically investigate and characterize the most discriminative forensic cues present in the temporal domain. A deeper understanding of such artifacts could lead to detectors that more effectively exploit spatial and temporal traces. Furthermore, our method was trained using fake videos generated by the Pyramid Flow model, selected for its ability to produce high-quality synthetic content with minimal perceptible visual artifacts. This design choice allowed us to focus on subtle, low-level traces rather than easily detectable flaws. However, as new generative models emerge, potentially relying on entirely different synthesis paradigms, the current approach may struggle to generalize. Although the core principles may still be applicable, the method itself may require adaptation or retraining.

## Acknowledgments and Disclosure of Funding

This work has received funding from the European Union under the Horizon Europe vera.ai project, Grant Agreement number 101070093, and was partially supported by SERICS (PE00000014) under the MUR National Recovery and Resilience Plan, funded by the European Union - NextGenerationEU. We thank David Luebke for early discussions on the project.

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

## Supplementary Material

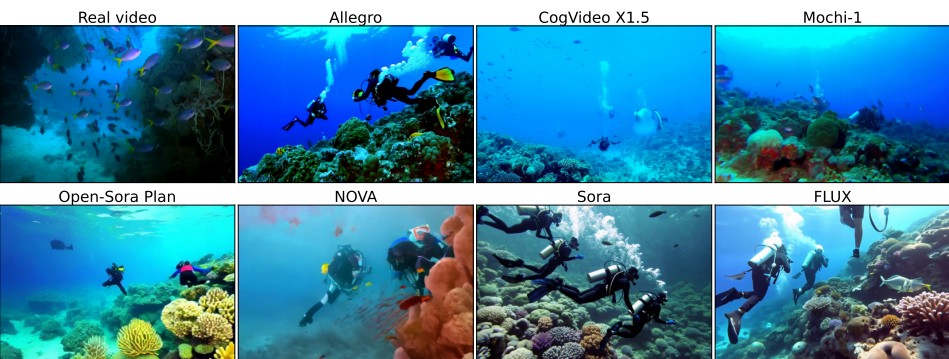

*"A group of scuba divers are swimming in a coral reef with colorful tropical fish."*

Figure 6: Examples of video from our proposed test dataset generated with recent state of the art generators.

## A   Dataset

To conduct our experiments, we created a dataset of synthetic videos generated using recent state-of-the-art models. For the real data, we relied on the Panda70M dataset [14], which provides access to YouTube video URLs, precise timestamps for captioned clips, accompanying captions and a relevance score indicating how well each clip aligns with its caption. We identify clips with favorable characteristics, excluding those that are screen recordings, contain static content, or exhibit minimal camera movement. To ensure a high-quality and diverse dataset, we selected only such type of clips, limiting the choice to a single clip per YouTube video. When multiple clips were available, we chose the one that was at least five seconds long and had the highest alignment score.

For our training, we selected $1,700$ real videos from the Panda70M dataset and used their captions to generate synthetic samples with one single generator Pyramid Flow [38] (version with 768p miniflux weights). We further augmented the synthetic set by applying the Pyramid Flow Variational Autoencoder (VAE) to the original real videos, resulting in $1,700$ more synthetic videos. The full set of $5,100$ videos was split into $4,200$ for training and $900$ for validation. For evaluation, we selected $300$ real videos and used their captions to generate synthetic counterparts using five recent video generation models: Allegro [95], Mochi-1 [73], CogVideoX [86] (with CogVideoX1.5-5B weights), NOVA [25], and Open-Sora Plan [45] (with 1.3B weights). To guarantee consistency with the real videos, we compressed all generated videos using exactly the same codec of the real videos, i.e. Advanced Video Coding (H.264). Specifically, we used the Main@L3.1 codec profile and a Constant Rate Factor (CRF) randomly sampled between 16 and 30.

Additionally, the same captions were used to produce other synthetic videos via two online platforms: Sora [8] and FLUX [29]. Overall, using 7 generators, we created $2,100$ synthetic videos with a frame-rate that goes from 12 fps (NOVA) to 30 fps (Mochi-1, Sora) and a spatial resolution that varies from $640 \times 352$ pixels (Open-Sora Plan) to $1360 \times 768$ pixels (CogVideoX). Some examples can be seen in Fig.6, Fig.10 and Fig.11.

## B   Artifacts analysis

In this Section we present an additional analysis of the low-level artifacts for other synthetic generators. In particular, in Fig.7, we show the close up of spatial and temporal-spatial power spectra before (top) and after (bottom) compression for additional synthetic generators. In all cases, we observe strong forensic artifacts (even for NOVA that relies on an autoregressive model) that are significantly reduced after compression. However, artifacts remain visible, particularly along the diagonal directions. It is also worth noting that temporal artifacts are less prominent, and the peaks along the vertical and horizontal directions tend to overlap with those introduced by the compression algorithm.

To further support our conjecture that the mid-high frequencies along the diagonal directions are more discriminative, we compare real and reconstructed videos in the frequency domain. Reconstructed

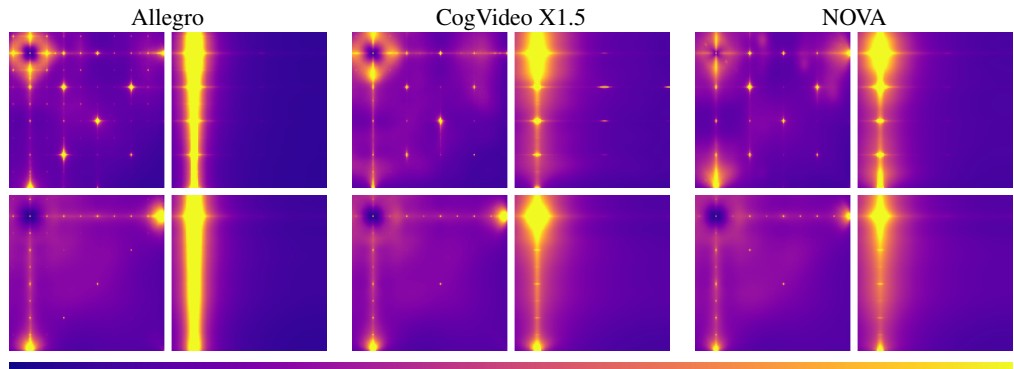

Figure 7: For each generator (Allegro, CogVideo X1.5, NOVA), we show the close up of its spatial power spectrum $S_{yx}(u,v)$ (left), and the close up of its temporal-spatial power spectra $S_{yt}(u,w)$ (right), both before compression (top) and after compression (bottom),

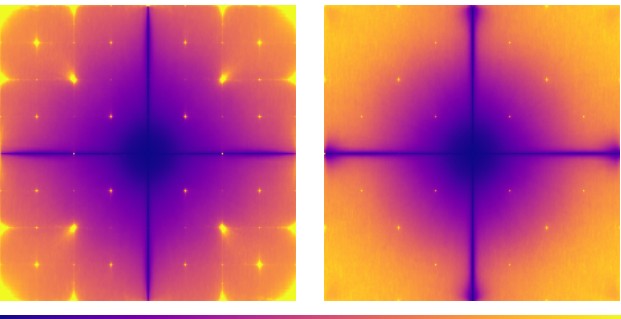

Figure 8: Average distance in the frequency domain as defined in eq.7, before compression (left) and after compression (right). The distance is computed between the real and the reconstructed video generated by the autoencoder of Pyramid Flow [38]. In this way we can show the artifacts of the model that are not influenced by the semantic content. The distance is nearly zero at the low frequencies and along the vertical and horizontal directions, much larger at mid-high frequencies along diagonal directions, even when videos are compressed.

videos are obtained with the same autoencoder used during generation (see Section 3.2), hence they share the same content as real videos, but embed the traces related to the generation architecture. Such traces can be highlighted by computing the following distance:

$$d(u,v) = \frac{1}{N} \sum_{i=1}^{N} \frac{\sum_w \left| X_i(u,v,w) - \hat{X}_i(u,v,w) \right|^2}{\sum_w |X_i(u,v,w)|^2} \tag{7}$$

where $X_i(u,v,w)$ and $\hat{X}_i(u,v,w)$ are the 3d-Fourier transforms of $i$-th real video and its reconstruction, respectively, while $N$ is the number of videos, that is equal to 100 in this experiment. In this comparison, we take into account the fact that small differences in low-energy regions of the spectrum are more significant than the same differences in high-energy regions. Therefore, we normalize the sum of squared differences by the energy of the real data at each spatial frequency. We show the average distance $d(u,v)$ on the left side of Fig.8, while the right side displays the same quantity evaluated after compressing the reconstructed videos using the same codec as for the real ones. Before compression (Fig.8, left) the reconstructed video shows a significant deviation from the real video at the mid-high frequencies, while horizontal and vertical frequencies appear to be less useful for detection. After compression (Fig. 8, right), the distances are reduced, but the diagonal mid-high frequency components are still discriminative.

To verify that the proposed detector leverages these mid-high frequencies to establish if a video is fake or not, we conducted a suitable toy experiment: at test time we modified the real videos and injected the mid-high frequencies of fully synthetic videos into them. We observe a performance drop on this test set. In particular, the True Negative Rate (TNR), which is the probability that real videos are correctly classified as real, goes from 98.7 to 0.2 (on average). This clearly shows that the detector inverts its decision.

Table 6: Performance by varying $L$ (number of wavelet decomposition levels).

| | AUC ↑ / bAcc ↑ | | | | | NLL ↓ / ECE ↓ | | | | |
|---|---|---|---|---|---|---|---|---|---|---|
| $L$ | Allegro | CogVideo X1.5 | Mochi-1 | OpenSora-Plan | AVG | Allegro | CogVideo X1.5 | Mochi-1 | OpenSora-Plan | AVG |
| 2 | **98.6** / 90.0 | **96.5** / 81.7 | **98.9** / 91.5 | **99.8** / 96.0 | **98.4** / 89.8 | 0.29 / .094 | 0.60 / .174 | 0.24 / .079 | 0.08 / .031 | 0.30 / .095 |
| 3 | **98.6** / **91.5** | **97.2** / **85.0** | **99.1** / **93.5** | **99.8** / **97.0** | **98.7** / **91.8** | **0.28** / **.076** | **0.53** / **.140** | **0.19** / **.056** | **0.07** / **.023** | **0.27** / **.074** |
| 4 | 97.6 / 88.8 | 95.7 / 81.5 | 98.2 / 89.3 | 99.1 / 94.5 | 97.7 / 88.5 | 0.39 / .102 | 0.67 / .168 | 0.31 / .091 | 0.18 / .045 | 0.39 / .101 |

Table 7: Comparison in terms of AUC with SoTA methods both trained on their original datasets and re-trained using Pyramid Flow (*). Bold underlines the best performance for each column with a margin of 1%.

| | Recent Generators (2024-25 years) | | | | | | | GenVideo (2022-23 years) | | | | | | | | |
|---|---|---|---|---|---|---|---|---|---|---|---|---|---|---|---|---|
| AUC ↑ | Allegro | CogV. X1.5 | OSora Plane | Mochi1 | Nova | Sora | Flux | Crafter | Gen2 | Hot Shot | LaVie | Model Scope | Morph Stu. | Show1 | Moon valley | AVG |
| DMID | 93.2 | 93.2 | 96.4 | 78.8 | 94.1 | 98.3 | **99.8** | **100.** | **99.7** | 98.9 | **99.9** | 99.2 | 99.6 | 99.7 | 99.9 | 96.7 |
| UnivFD | 47.7 | 60.1 | 42.8 | 53.3 | 73.9 | 32.4 | 40.6 | 88.0 | 74.8 | 77.5 | 85.3 | 95.0 | 86.3 | 76.4 | 79.6 | 67.6 |
| RINE | 83.3 | 82.7 | 92.2 | 82.0 | 11.0 | 0.7 | 1.0 | **99.9** | **99.5** | 95.4 | **99.5** | 99.0 | 99.3 | 99.6 | 99.4 | 76.3 |
| FreqNet | 75.1 | 43.1 | 80.5 | 52.8 | 37.3 | 56.9 | 39.8 | 57.4 | 30.8 | 23.2 | 55.7 | 42.5 | 54.2 | 48.7 | 58.1 | 50.4 |
| FIRE | 56.0 | 51.2 | 93.0 | 85.8 | 79.7 | 40.8 | 46.8 | 36.9 | 23.0 | 35.1 | 62.0 | 25.7 | 24.3 | 40.1 | 18.6 | 47.9 |
| DMID* | **99.5** | **98.7** | **99.9** | 96.6 | **99.4** | 99.3 | **99.8** | 98.8 | **99.8** | 93.3 | 97.6 | 96.9 | 97.9 | 97.2 | **99.6** | 98.3 |
| UnivFD* | 93.5 | 83.6 | 92.8 | 89.3 | 93.5 | 97.3 | 94.4 | 97.2 | 98.0 | 81.7 | 94.0 | 91.7 | 94.6 | 90.6 | 98.9 | 92.7 |
| RINE* | **99.7** | 92.3 | **99.1** | 96.8 | **98.6** | 97.5 | 97.1 | 98.6 | 98.9 | 90.7 | 97.5 | 97.0 | 97.5 | 93.8 | **99.7** | 97.0 |
| FreqNet* | 91.6 | 72.3 | 94.3 | 84.3 | 79.2 | 73.7 | 69.6 | 53.5 | 63.6 | 41.1 | 76.0 | 42.4 | 59.8 | 69.2 | 71.8 | 69.5 |
| FIRE* | 83.9 | 68.6 | 66.4 | 79.8 | 83.4 | 70.3 | 80.8 | 92.2 | 95.0 | 84.3 | 85.1 | 84.4 | 91.0 | 83.2 | 97.4 | 83.0 |
| AIGVDet | 80.7 | 76.0 | 85.4 | 78.5 | 88.0 | 89.4 | 89.1 | 98.0 | 95.9 | 97.0 | 96.4 | 73.5 | 94.0 | 80.6 | **99.9** | 88.2 |
| DeMamba | 78.7 | 97.1 | 95.6 | 86.6 | 82.8 | 65.8 | 60.5 | **100.** | **100.** | 99.6 | 99.7 | 94.7 | **100.** | 99.9 | **100.** | 90.7 |
| MM-Det | 53.9 | 0.0 | 47.7 | 44.3 | 41.1 | 44.8 | 53.8 | 46.5 | 43.1 | 0.0 | 49.7 | 31.9 | 49.6 | 51.0 | 37.9 | 39.7 |
| AIGVDet* | 92.3 | 91.2 | 42.4 | 91.4 | 94.8 | 88.7 | 91.4 | **99.9** | **100.** | 98.6 | 95.0 | **99.3** | 99.9 | 88.2 | **100.** | 91.5 |
| DeMamba* | 97.9 | 88.5 | **99.2** | 93.8 | 90.8 | 94.2 | 95.3 | 94.7 | 97.2 | 70.0 | 92.3 | 78.2 | 92.9 | 85.7 | 96.3 | 91.1 |
| MM-Det* | 89.2 | 0.0 | 97.2 | 96.5 | 97.3 | 96.6 | 97.0 | 93.3 | 95.4 | 0.0 | 94.3 | 69.6 | 84.5 | 89.1 | 95.0 | 79.7 |
| Ours | 98.6 | 97.2 | **99.8** | 99.1 | 99.3 | 99.9 | 99.9 | 99.8 | **100.** | 91.0 | 97.7 | **99.4** | 99.3 | 97.1 | **99.9** | 98.5 |

# C Additional results

**Ablation.** In Table 6, We conducted a study to assess the impact of the number of decomposition levels of the wavelet transform implemented in our augmentation strategy. Specifically, we evaluated configurations with 2, 3, and 4 levels of decomposition. The results indicate that the choice of 3 decomposition levels yields the better performance across multiple metrics. The difference is more marked if we look at the Accuracy than the AUC. Even NLL and especially ECE show an advantage in this situation, which confirms that this choice helps to achieve more calibrated results.

**SoTA comparison.** In Table 1 and 2 of the main paper, we report the comparison with SoTA methods only in terms of balanced Accuracy. Here, in Tables 7, 8, 9 and 10, we show the performance using the other four metrics: AUC, Pd@5, NLL and ECE. AUC is generally higher than the balanced Accuracy for most of the methods, with variations even superior to 15%. This means that a fixed threshold set to 0.5 is not the best choice and proper calibration is needed. However, our approach is instead able to achieve good performance consistently over all the metrics, in particular it gains an improvement on average of around 53% and 68% for NLL and ECE, respectively.

**Compression robustness.** We compressed the test dataset used in our ablation study with H.264 at different quality levels by varying the Constant Rate Factor (CRF). Table 11 reports results across decomposition levels, showing robustness to compression (AUC always above 80%). Accuracy drops under severe compression (around 60% at CRF 32) motivating a threshold calibration procedure for low-quality compressed videos (calibrating with only 10 compressed validation videos raises accuracy to 76%).

Table 11: Performance by varying compression quality (CRF) and the number of wavelet decomposition levels ($L$)).

| AUC ↑ / bAcc ↑ | $L$=2 | $L$=3 | $L$=4 |
|---|---|---|---|
| CRF 20 (high quality) | 97.8 / 83.8 | 97.3 / 84.5 | 96.6 / 83.2 |
| CRF 24 | 96.8 / 77.1 | 96.0 / 78.3 | 95.2 / 78.2 |
| CRF 28 | 93.9 / 67.9 | 92.5 / 69.1 | 91.1 / 68.5 |
| CRF 32 (low quality) | 88.2 / 59.5 | 85.2 / 59.2 | 82.6 / 59.3 |

Table 8: Comparison in terms of Pd@5 with SoTA methods both trained on their original datasets and re-trained using Pyramid Flow (*). Bold underlines the best performance for each column with a margin of 1%.

| Pd@5 ↑ | Allegro | CogV. X1.5 | OSora Plane | Mochi1 | Nova | Sora | Flux | Crafter | Gen2 | Hot Shot | LaVie | Model Scope | Morph Stu. | Show1 | Moon valley | AVG |
|---|---|---|---|---|---|---|---|---|---|---|---|---|---|---|---|---|
| | | | | | | | | | **Recent Generators (2024-25 years)** → **GenVideo (2022-23 years)** | | | | | | | |
| DMID | 73.0 | 76.0 | 82.7 | 40.0 | 72.3 | 92.3 | 99.9 | **100.** | 99.0 | 96.0 | **99.9** | 96.6 | 99.0 | 99.1 | 100. | 88.4 |
| UnivFD | 3.7 | 10.3 | 4.3 | 7.3 | 26.0 | 1.7 | 3.0 | 59.6 | 31.4 | 40.7 | 52.7 | 77.6 | 52.9 | 35.9 | 40.6 | 29.8 |
| RINE | 44.3 | 39.3 | 73.7 | 46.0 | 8.3 | 0.7 | 0.7 | 99.9 | 98.7 | 77.4 | 98.5 | **97.7** | 98.0 | 98.1 | 99.0 | 65.4 |
| FreqNet | 16.7 | 4.8 | 22.2 | 5.6 | 0.9 | 7.5 | 3.0 | 14.8 | 3.6 | 4.4 | 12.8 | 8.1 | 12.7 | 9.6 | 12.4 | 9.3 |
| FIRE | 6.3 | 5.7 | 84.7 | 70.3 | 62.3 | 1.3 | 2.0 | 0.0 | 0.0 | 0.0 | 1.6 | 0.6 | 0.0 | 0.0 | 0.0 | 15.7 |
| DMID* | 97.3 | **94.3** | **99.0** | 87.0 | **97.3** | 98.0 | 98.7 | 95.9 | **99.7** | 81.0 | 90.9 | 88.9 | 90.3 | 85.9 | **99.2** | 93.6 |
| UnivFD* | 67.3 | 45.7 | 67.7 | 61.0 | 68.3 | 86.3 | 75.3 | 85.8 | 90.9 | 38.7 | 75.2 | 66.4 | 76.4 | 64.4 | 94.9 | 71.0 |
| RINE* | **98.3** | 67.3 | 95.7 | 84.3 | 93.3 | 89.0 | 87.0 | 94.7 | 95.4 | 64.1 | 87.6 | 84.9 | 86.4 | 73.0 | 98.6 | 86.7 |
| FreqNet* | 56.7 | 22.7 | 71.7 | 33.7 | 18.3 | 11.3 | 5.7 | 6.7 | 13.2 | 6.6 | 33.3 | 3.7 | 11.0 | 16.1 | 13.7 | 21.6 |
| FIRE* | 45.7 | 26.3 | 28.7 | 50.7 | 51.3 | 22.7 | 44.0 | 67.3 | 70.0 | 45.9 | 48.9 | 38.9 | 59.0 | 49.3 | 88.5 | 49.1 |
| AIGVDet | 34.7 | 29.0 | 49.3 | 40.7 | 51.3 | 71.7 | 67.7 | 94.1 | 87.3 | 91.1 | 88.3 | 44.1 | 80.7 | 57.1 | **99.8** | 65.8 |
| DeMamba | 21.3 | 88.7 | 81.0 | 62.3 | 50.0 | 22.0 | 22.7 | **100.** | 99.9 | 98.7 | 99.0 | 71.4 | **100.** | 99.4 | 100. | 74.4 |
| MM-Det | 5.6 | 0.0 | 4.6 | 4.3 | 3.8 | 4.3 | 5.5 | 4.6 | 4.3 | 0.0 | 5.0 | 3.1 | 5.0 | 5.1 | 3.8 | 3.9 |
| AIGVDet* | 71.7 | 59.0 | 23.3 | 68.7 | 72.0 | 64.7 | 65.3 | 99.4 | 99.9 | 94.4 | 75.8 | 96.1 | **99.7** | 43.9 | 100. | 75.6 |
| DeMamba* | 94.0 | 59.7 | 97.3 | 82.0 | 69.7 | 78.7 | 85.7 | 81.8 | 90.1 | 31.1 | 69.9 | 45.3 | 75.3 | 56.0 | 89.3 | 73.7 |
| MM-Det* | 52.3 | 0.0 | 85.7 | 83.7 | 84.7 | 85.7 | 82.7 | 72.2 | 76.2 | 0.0 | 74.1 | 25.1 | 39.1 | 52.6 | 76.7 | 59.4 |
| Ours | 93.0 | 84.3 | **99.0** | 94.7 | 95.3 | **100.** | **99.7** | 99.1 | 99.9 | 72.9 | 94.3 | **97.9** | 97.3 | 91.0 | **99.5** | 94.5 |

Table 9: Comparison in terms of NLL with SoTA methods both trained on their original datasets and re-trained using Pyramid Flow (*). Bold underlines the best performance for each column with a margin of 1%.

| NLL ↓ | Allegro | CogV. X1.5 | OSora Plane | Mochi1 | Nova | Sora | Flux | Crafter | Gen2 | Hot Shot | LaVie | Model Scope | Morph Stu. | Show1 | Moon valley | AVG |
|---|---|---|---|---|---|---|---|---|---|---|---|---|---|---|---|---|
| DMID | 2.18 | 2.14 | 1.53 | 3.97 | 2.10 | 1.10 | 0.73 | **0.04** | 0.92 | 2.09 | 0.20 | 1.82 | 1.17 | 1.10 | 1.08 | 1.48 |
| UnivFD | 2.76 | 2.16 | 3.02 | 2.50 | 1.45 | 3.62 | 3.13 | 1.76 | 2.67 | 2.44 | 1.96 | 1.21 | 1.93 | 2.56 | 2.34 | 2.37 |
| RINE | 0.59 | 0.60 | 0.45 | 0.59 | 6.62 | 17.2 | 15.6 | 0.14 | 0.30 | 0.50 | 0.16 | 0.36 | 0.32 | 0.26 | 0.34 | 2.93 |
| FreqNet | 10.6 | 15.6 | 10.2 | 13.6 | 16.8 | 13.7 | 16.8 | 30.5 | 33.0 | 35.5 | 30.6 | 33.6 | 30.7 | 30.7 | 29.5 | 23.4 |
| FIRE | 3.95 | 4.01 | 3.94 | 3.94 | 3.95 | 3.95 | 3.95 | 4.84 | 4.84 | 5.00 | 4.86 | 5.01 | 4.85 | 4.96 | 4.84 | 4.46 |
| DMID* | 0.41 | 0.51 | 0.19 | 0.57 | 0.35 | 0.26 | 0.22 | 0.20 | 0.14 | 0.69 | 0.32 | 0.50 | 0.41 | 0.54 | 0.14 | 0.36 |
| UnivFD* | 0.53 | 1.00 | 0.56 | 0.71 | 0.54 | 0.27 | 0.46 | 0.22 | 0.19 | 0.88 | 0.37 | 0.48 | 0.35 | 0.52 | 0.14 | 0.48 |
| RINE* | **0.24** | 2.61 | 0.42 | 1.27 | 0.68 | 0.95 | 1.11 | 0.24 | 0.19 | 1.46 | 0.38 | 0.45 | 0.38 | 0.97 | 0.07 | 0.76 |
| FreqNet* | 0.65 | 1.80 | 0.61 | 1.02 | 1.17 | 1.52 | 1.66 | 2.93 | 2.34 | 4.66 | 2.35 | 4.31 | 2.76 | 2.27 | 1.96 | 2.13 |
| FIRE* | 0.51 | 0.77 | 0.82 | 0.62 | 0.54 | 0.70 | 0.56 | 0.39 | 0.29 | 0.66 | 0.63 | 0.65 | 0.43 | 0.68 | 0.20 | 0.56 |
| AIGVDet | 0.95 | 1.16 | 0.84 | 1.07 | 0.67 | 0.70 | 0.69 | 0.31 | 0.46 | 0.48 | 0.52 | 1.35 | 0.46 | 1.12 | 0.32 | 0.74 |
| DeMamba | 1.98 | **0.33** | 0.48 | 1.12 | 1.41 | 2.57 | 2.78 | 0.05 | 0.05 | **0.08** | **0.07** | 0.64 | **0.05** | **0.06** | **0.05** | 0.78 |
| MM-Det | 38.4 | 88.3 | 38.4 | 38.6 | 38.5 | 38.6 | 38.4 | 48.3 | 48.3 | 98.3 | 48.3 | 48.8 | 48.3 | 48.3 | 48.4 | 50.4 |
| AIGVDet* | 0.60 | 0.72 | 3.57 | 0.64 | 0.56 | 0.77 | 0.69 | 0.44 | 0.30 | 2.72 | 4.38 | 1.08 | 0.56 | 7.55 | 0.35 | 1.66 |
| DeMamba* | 0.49 | 1.66 | 0.22 | 0.85 | 1.31 | 1.00 | 0.83 | 0.45 | 0.26 | 1.93 | 0.67 | 1.47 | 0.58 | 1.08 | 0.30 | 0.87 |
| MM-Det* | 3.60 | 50.0 | 1.50 | 1.71 | 1.62 | 1.46 | 1.55 | 0.74 | 0.63 | 50.5 | 0.69 | 2.13 | 1.13 | 0.89 | 0.65 | 7.92 |
| Ours | 0.28 | 0.53 | **0.07** | **0.19** | **0.18** | **0.04** | **0.05** | 0.06 | **0.05** | 0.53 | 0.16 | **0.09** | 0.09 | 0.22 | 0.05 | **0.17** |

**Training data.** In this paragraph, we evaluate the influence of training data. In Table 12, the proposed augmentation yields clear gains even with limited data (20% of the training set) and when training on a different generator (CogVideoX instead of Pyramid Flow). With only 20% of the data, augmentation still improves balanced accuracy (+3.2%). Likewise, when replacing Pyramid Flow with CogVideoX, we again observe a substantial boost from the proposed augmentation strategy (+6.3%).

Table 12: Performance by varying the type of generator and the size of the dataset during training.

| training data | aug. | Allegro | Mochi-1 | OpenSora-Plan | AVG |
|---|---|---|---|---|---|
| | | | | **AUC ↑ / bAcc ↑** | |
| 100% Pyr. Flow | | 94.7 / 68.2 | 94.9 / 75.5 | 98.0 / 80.0 | 95.9 / 74.6 |
| 100% Pyr. Flow | ✓ | 98.6 / 91.5 | 99.1 / 93.5 | 99.8 / 97.0 | 99.1 / 94.0 |
| 20% Pyr. Flow | | 94.3 / 70.8 | 93.9 / 77.8 | 98.4 / 87.3 | 95.5 / 78.7 |
| 20% Pyr. Flow | ✓ | 97.6 / 88.3 | 98.8 / 92.7 | 99.8 / 96.5 | 98.7 / 92.5 |
| 100% CogV. X1.5 | | 90.7 / 69.3 | 90.7 / 72.0 | 94.9 / 78.0 | 92.1 / 73.1 |
| 100% CogV. X1.5 | ✓ | 98.5 / 88.8 | 97.6 / 85.2 | 99.2 / 92.8 | 98.4 / 88.9 |

Table 10: Comparison in terms of ECE with SoTA methods both trained on their original datasets and re-trained using Pyramid Flow (*). Bold underlines the best performance for each column with a margin of 1%.

| ECE ↓ | Recent Generators (2024-25 years) | | | | | | | GenVideo (2022-23 years) | | | | | | | | AVG |
|---|---|---|---|---|---|---|---|---|---|---|---|---|---|---|---|---|
| | Allegro | CogV. X1.5 | OSora Plane | Mochi1 | Nova | Sora | Flux | Crafter | Gen2 | Hot Shot | LaVie | Model Scope | Morph Stu. | Show1 | Moon valley | |
| DMID | .406 | .406 | .312 | .471 | .399 | .334 | .329 | .026 | .274 | .436 | .068 | .362 | .289 | .347 | .341 | .320 |
| UnivFD | .479 | .419 | .481 | .438 | .324 | .501 | .489 | .398 | .464 | .445 | .410 | .348 | .420 | .458 | .440 | .434 |
| RINE | .152 | .149 | .167 | .134 | .536 | .623 | .624 | .107 | .202 | .229 | .107 | .230 | .208 | .173 | .230 | .258 |
| FreqNet | .238 | .449 | .217 | .381 | .497 | .361 | .470 | .180 | .286 | .378 | .184 | .281 | .194 | .198 | .153 | .298 |
| FIRE | .483 | .492 | .482 | .482 | .482 | .484 | .482 | .503 | .505 | .520 | .509 | .529 | .507 | .522 | .506 | .499 |
| DMID* | .227 | .254 | .125 | .237 | .189 | .137 | .140 | .078 | .086 | .241 | .126 | .220 | .193 | .248 | .080 | .172 |
| UnivFD* | .195 | .299 | .200 | .229 | .206 | .094 | .162 | .039 | .024 | .252 | .096 | .142 | .091 | .151 | .024 | .147 |
| RINE* | .098 | .339 | .110 | .225 | .172 | .190 | .198 | .032 | .023 | .202 | .075 | .103 | .079 | .162 | **.010** | .134 |
| FreqNet* | .108 | .296 | .106 | .170 | .244 | .289 | .361 | .404 | .347 | .512 | .312 | .499 | .368 | .309 | .289 | .308 |
| FIRE* | .086 | .156 | .171 | .129 | .100 | .144 | .091 | .069 | .051 | .158 | .141 | .168 | .090 | .142 | .044 | .116 |
| AIGVDet | .287 | .305 | .264 | .283 | .233 | .204 | .206 | .157 | .192 | .239 | .230 | .293 | .150 | .277 | .230 | .237 |
| DeMamba | .410 | **.079** | .120 | .211 | .274 | .415 | .405 | .020 | **.019** | **.012** | **.014** | .171 | .020 | **.014** | .019 | .147 |
| MM-Det | .129 | .612 | .126 | .140 | .139 | .143 | .129 | **.019** | .025 | .518 | .018 | .070 | .019 | .018 | .034 | .143 |
| AIGVDet* | .230 | .292 | .411 | .233 | .243 | .266 | .258 | .181 | .162 | .420 | .442 | .305 | .248 | .492 | .250 | .296 |
| DeMamba* | .159 | .364 | .074 | .193 | .291 | .249 | .224 | .092 | .047 | .346 | .167 | .278 | .127 | .236 | .050 | .193 |
| MM-Det* | .421 | .502 | .272 | .284 | .292 | .239 | .268 | .111 | .089 | .593 | .101 | .305 | .170 | .134 | .091 | .258 |
| Ours | **.076** | .140 | **.023** | **.056** | **.058** | **.014** | **.012** | .025 | .029 | .145 | .016 | **.013** | **.012** | .043 | .026 | **.046** |

# D   Implementations details

The proposed solution uses as backbone the Vision Transformer (ViT) architecture with 4 registers [22] and a pre-training based on DINOv2 strategy [57]. Specifically for the model, we use the implementation available in the PyTorch Image Models (TIMM) library followed by a single fully connected layer for binary classification. The model processes an input of $36 \times 36$ patches, each of size $14 \times 14$ pixels. Training is performed using the cross-entropy loss function and the ADAM optimizer, with a learning rate of $10^{-6}$, a weight decay of $10^{-6}$, and a batch size of 22. During training, we apply the augmentation strategy described in the main paper with a probability of $10\%$. Frames are randomly cropped to match the input size of the network, and each batch contains an equal number of real and fake crops. We evaluate the balanced accuracy on the validation set every $2,035$ iterations. Training is conducted for 36 evaluation steps, and only the model weights corresponding to the best validation performance are kept and used for testing. At inference, the prediction pipeline consists of the following steps (Fig. 9):

1. the first 64 frames of the video are extracted;
2. the frame is center-cropped to match the network input size;
3. the network processes the frame and produces a corresponding logit value;
4. video-level prediction is computed by averaging the 64 logit values.

Finally, we report the computational times. The analysis is carried out on a hardware with two CPUs Intel(R) Xeon(R) Gold 5220 CPU @ 2.20GHz with 128GB of RAM, and a Nvidia A6000 GPU with 48GB of VRAM. The proposed method exhibits a relatively low inference time: it takes around 13 seconds to analyze a video of 64 frames, which is comparable with MM-Det and lower than DeMamba (21 sec) and AIGVDet (180 sec). The total training time for our method is about 77 hours.

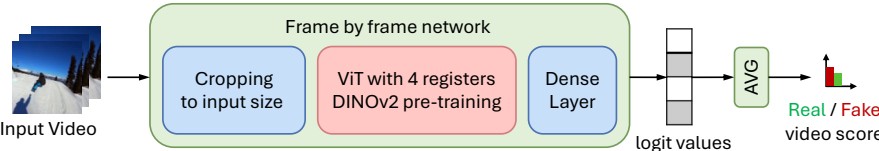

Figure 9: Prediction pipeline. At inference, we use up to 64 frames per video, center-cropped to match the model input. Video-level predictions are computed by averaging logits across frames.

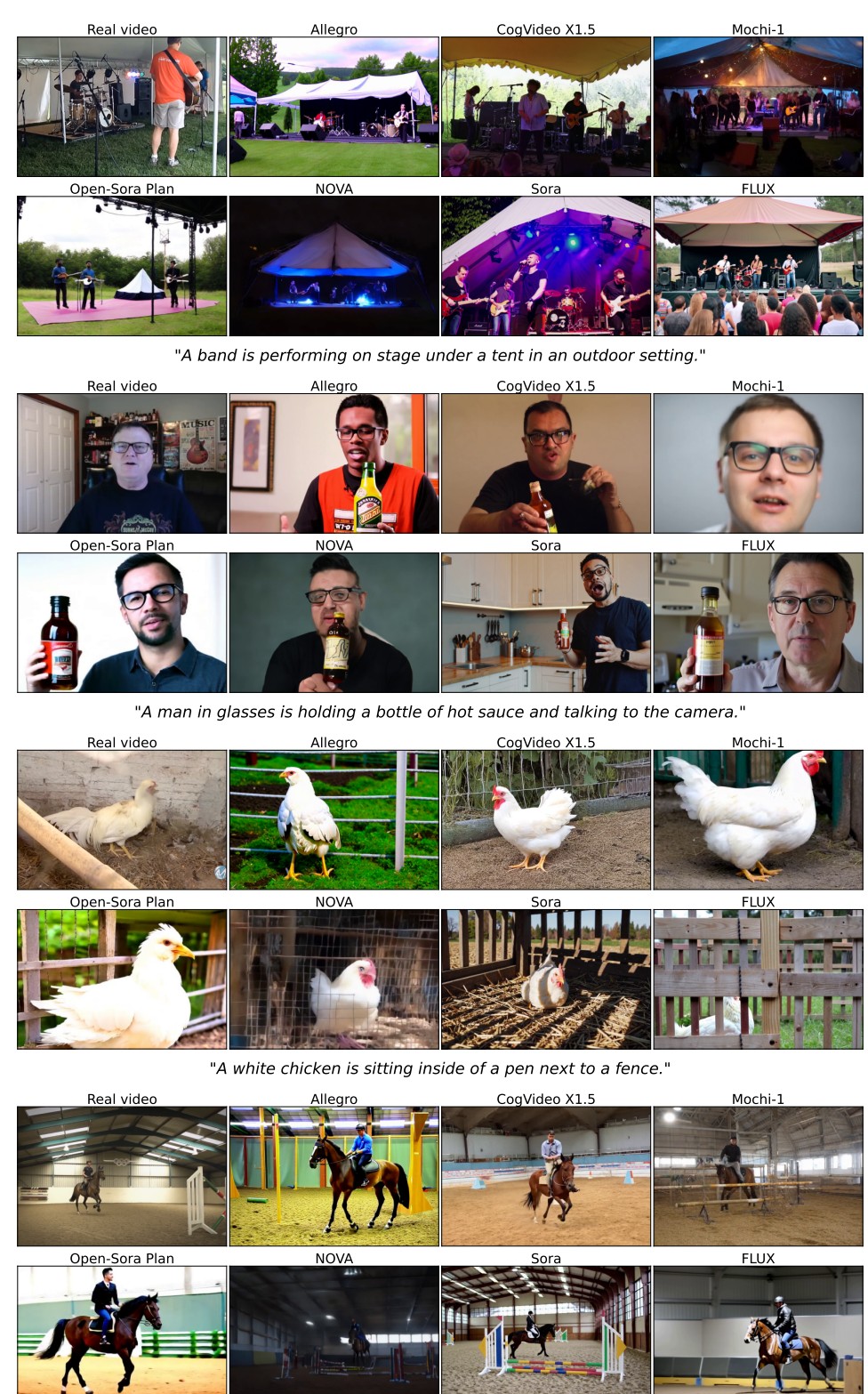

Figure 10: Some examples of videos from our dataset that we created with recent state of the art generators.

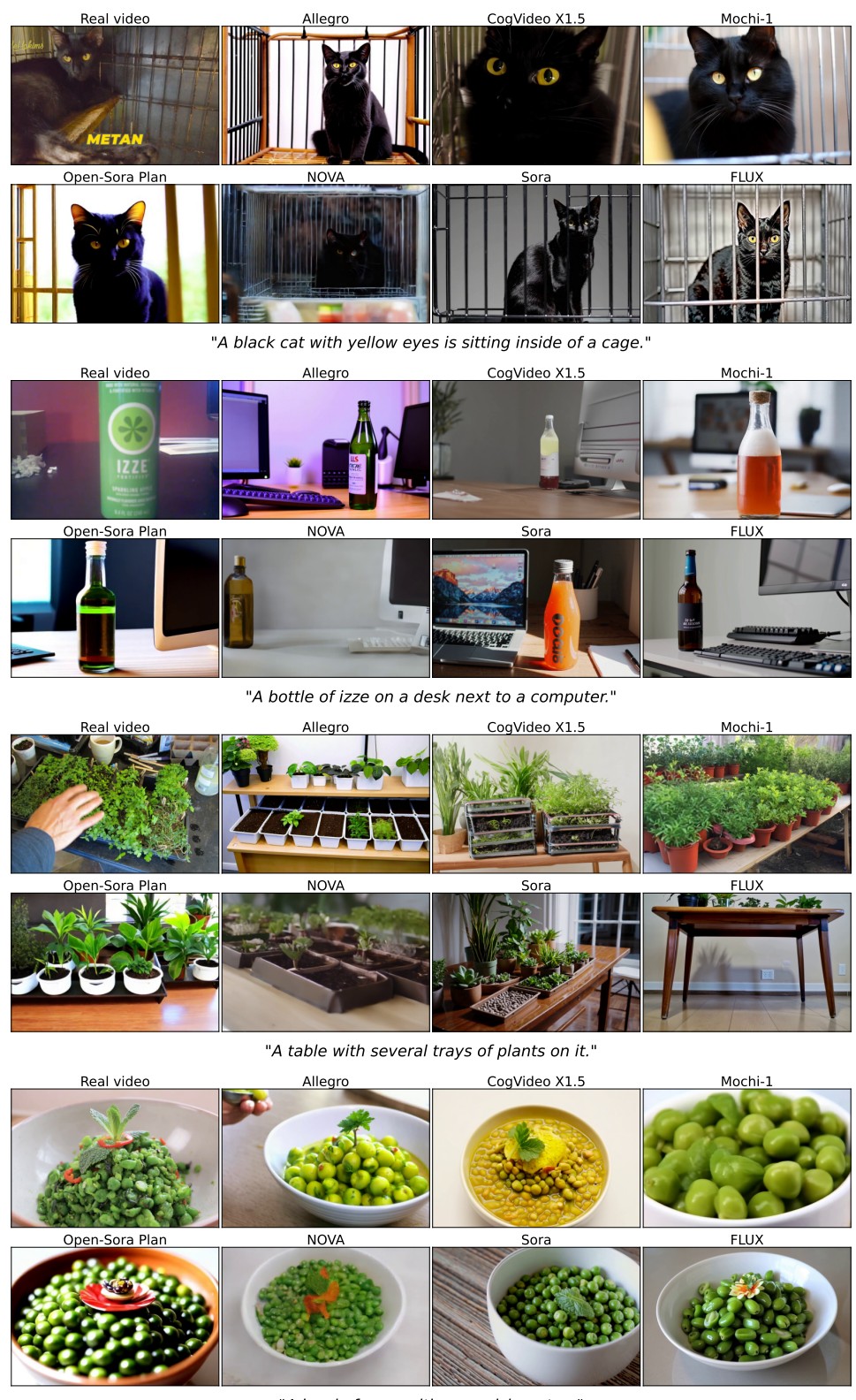

Figure 11: Some examples of videos from our dataset that we created with recent state of the art generators.

