# OpenReview forum: "Seeing What Matters: Generalizable AI-generated Video Detection with Forensic-Oriented Augmentation"
_NeurIPS.cc/2025/Conference — NeurIPS 2025 poster_

### Official Review · Reviewer_Nt42 · 2025-06-16

**Clarity:** 3
**Significance:** 4
**Originality:** 2
**Rating:** 5
**Confidence:** 3

**Summary:**

The submission proposes a combination of two data augmentation strategies relying on a very recent video encoder (Pyramid Flow, ICLR 2025) for the detection of ai-generated videos.

The first data augmentation strategy (Section 3.2) relies on encoding real videos in an autoencoder fashion.
The second data augmentation strategy (Section 3.2, fig 4) runs a Haar transform on an AI-generated video, embeds for the low band the frequencies taken from real videos and inverts it to create an altered AI-generated video. This aims to force the method to look at mid to high frequencies.

They observe that several SOTA methods which perform well on videos from generators of 2022-2023 perform worse on videos from generators of 2024-now.

They show good performance for video detection with their method.  They show that retraining the other SOTA detectors does not close the gap to their proposed method fully.

They show that one image-based method improves substantially and gets close to the proposed method when retrained using the encoder from Pyramid Flow.

More Detailed technical descriptions for the implementation can be found in the supplement section D.

**Questions:**

- Can you place a link to section D of the appendix in the main paper in section 4 ?

- A graphic with the full prediction model (incl link to the backbones discussed) in the supplement would be helpful .  That would help to consolidate the information which is a bit scattered in  the main paper.

**Ethical Concerns:**

["NO or VERY MINOR ethics concerns only"]

**Final Justification:**

Good paper, maybe an oral.

**Limitations:**

They discussed the limitations meaningfully.

**Quality:**

4

**Strengths And Weaknesses:**

Strengths:
- easy to read
- simple idea
- training details in the supplement are available

- ablation study that attempts to use the proposed encoder with other near-SOTA methods
- ablation study on data augmentation
- ablation study on backbones

Weaknesses:
- somewhat limited novelty

---

> ### Author Rebuttal · Authors · 2025-07-30
>
> We thank the reviewer for their positive feedback. We are glad that they found our paper “easy to read”, with a “simple idea” and containing an “extensive ablation study”.
>
> In the following, we address each question individually.
>
> **Q1: Can you place a link to section D of the appendix in the main paper in section 4?**
>
> Thank you for the suggestion! Yes, we will include this link in the paper and better connect the main paper to the appendix.
>
> **Q2: A graphic with the full prediction model (incl link to the backbones discussed) in the supplement would be helpful. That would help to consolidate the information which is a bit scattered in the main paper.**
>
> Yes, we will add such a graphic in the supplemental material and in the main paper, and describe more clearly the information about the models used.

---

### Official Review · Reviewer_5ZLP · 2025-06-30

**Clarity:** 4
**Significance:** 4
**Originality:** 4
**Rating:** 5
**Confidence:** 5

**Summary:**

This paper addresses the critical challenge of generalizing AI-generated video detection by proposing a novel forensic-oriented data augmentation strategy. Recognizing that existing detectors often struggle due to reliance on high-level, generator-specific semantic artifacts, the authors shift the focus towards intrinsic low-level, frequency-domain traces that are consistent across various generative architectures. Specifically, the paper introduces a wavelet-based augmentation strategy that selectively emphasizes mid-high diagonal frequency components robust to video compression artifacts. The proposed method is evaluated using videos from multiple generative models, achieving substantial improvements over current state-of-the-art methods, especially in cross-generator generalization scenarios.

**Questions:**

1.Temporal Artifacts: Given the promising results, how could the proposed augmentation strategy be extended or adapted to explicitly include temporal forensic artifacts, which could further enhance detection performance?
2.Future Generators: How robust do you expect your method to be against significantly different generative paradigms that may emerge in the future? What adaptations, if any, would be necessary to maintain high generalization?
3.Computational Efficiency: While wavelet decomposition is mentioned as computationally efficient, could you provide specific benchmarking or computational complexity analyses comparing this augmentation strategy to simpler augmentations such as MixUp or CutMix?
4.Dataset Diversity: Do you anticipate similar performance benefits when applying your augmentation strategy to significantly smaller or differently structured training datasets?

**Ethical Concerns:**

["NO or VERY MINOR ethics concerns only"]

**Final Justification:**

Tthe authors responded to all the comments, the overall quality of the paper is good.

**Limitations:**

Yes, the authors clearly and effectively addressed limitations and potential negative societal impacts. They acknowledge explicitly the method's limitation in focusing primarily on spatial artifacts and highlight the potential need for adaptation to new generative paradigms.

**Paper Formatting Concerns:**

No formatting issues were identified; the manuscript follows the NeurIPS guidelines effectively.

**Quality:**

4

**Strengths And Weaknesses:**

Strengths:
Quality:
1.The methodology is rigorous, clearly motivated by a thorough analysis of the frequency-domain forensic artifacts inherent in synthetic video generation.
2.Experiments are comprehensive and well-structured, using diverse datasets including recent generators and established benchmarks.
3.Metrics used (AUC, balanced accuracy, Pd@5%, NLL, ECE) are relevant and provide a nuanced evaluation.
Clarity:
1.The paper is very clearly written, with logical progression from motivation through method, results, and discussion.
2.Figures effectively illustrate the key insights about forensic artifacts and the proposed augmentation approach.
Significance:
1.The approach directly addresses a significant limitation in current forensic video detection research, offering practical strategies for real-world applicability.
2.Results demonstrate meaningful improvements in generalization, a critical aspect given the rapidly evolving nature of generative technologies.
Originality:
1.The wavelet-based augmentation strategy is novel within the context of forensic-oriented video detection.
2.The explicit focus on intrinsic, compression-resistant frequency bands is a valuable innovation in the forensic detection literature.


Weaknesses:
1.The method primarily focuses on spatial frequency components, explicitly noting a limitation in addressing temporal artifacts, which could be important in video detection tasks.
2.While the approach improves generalization substantially, the robustness of this augmentation strategy to drastically different future generative paradigms remains somewhat uncertain.

---

> ### Author Rebuttal · Authors · 2025-07-30
>
> We thank the reviewer for their positive feedback on our paper. We are glad that they found our methodology “rigorous, clearly motivated by a thorough analysis of the forensic artifacts”, our wavelet-based augmentation strategy “novel within the context of forensic-oriented video detection”, our experiments “comprehensive and well-structured, including recent generators and established benchmarks”, our paper “clearly written, with logical progression from motivation through method, results, and discussion”, that our “approach directly addresses a significant limitation in current forensic video detection research” and that our “results demonstrate meaningful improvements in generalization”.
>
> In the following, we address each question individually.
>
> **Q1: Temporal Artifacts: Given the promising results, how could the proposed augmentation strategy be extended or adapted to explicitly include temporal forensic artifacts, which could further enhance detection performance?**
>
> Our augmentation strategy is designed to prevent the detector from focusing on horizontal and vertical spatial frequencies where compression artifacts may overlap with forensic cues. Similarly, this approach can be extended to the temporal domain, encouraging the detector to exploit temporal forensic artifacts. One possible direction is to replace the low temporal frequencies of the fake video with those from the real video during training, while preserving the mid-to-high temporal frequencies of the fake content. We plan to take this direction in our future work as well as better understand the forensic artifacts in the temporal domain, since we believe that the replacement strategy should be focused on specific temporal frequencies. However, a key challenge in this respect is to identify a 3D backbone visual encoder capable of matching the performance of 2D models like DINOv2. As we can see from Table 2 the performance of the latter is significantly better than that of the 3D Hiera video model.
>
> **Q2: Future Generators: How robust do you expect your method to be against significantly different generative paradigms that may emerge in the future? What adaptations, if any, would be necessary to maintain high generalization?**
>
> The forensic artifacts that we have analyzed in our work are caused by the upsampling operation in the network’s  architecture (resulting in peaks in the frequency domain), which has also been shown in prior work [A,B]. This is caused by the inability of the generator to perfectly reproduce mid frequencies of the video signal as also shown in [C,D]. If new generative paradigms still include upsampling, then such traces will likely be present (maybe only in different positions). This is also what we observed with autoregressive models that are different from diffusion models. Despite having a different generative paradigm, they still present similar artifacts (see Figure 2 of the main paper). However, it is not easy to predict what will happen in the future. Hence if the generative paradigm will completely change, then the method would probably require re-training on a more recent generator. Even so, we believe that the discussion on the compression artifacts would still be relevant and hence our augmentation strategy could still be applied as is.
>
> References:
> [A] Zhang et al. Detecting and Simulating Artifacts in GAN Fake Images, WIFS 2019.
> [B] Vahdati et al. Beyond deepfake images: Detecting AI-generated videos, CVPRW 2024.
> [C] Dzanic et al. Fourier spectrum discrepancies in deep network generated images, NeurIPS 2020.
> [D] Corvi et al. Intriguing properties of synthetic images: from generative adversarial networks to diffusion models, CVPRW 2023.
>
> **Q3: Computational Efficiency: While wavelet decomposition is mentioned as computationally efficient, could you provide specific benchmarking or computational complexity analyses comparing this augmentation strategy to simpler augmentations such as MixUp or CutMix?**
>
> While wavelet decomposition is computationally efficient, it is slower than operations such as MixUp or CutMix. More specifically, the wavelet augmentation (with three decomposition levels) requires about 21*N FLOPs (floating-point operations), while MixUp requires only 3*N FLOPs, where N is the number of pixels. However, the implementation of the wavelet decomposition using convolution reduces the execution time substantially on parallel hardware architectures (GPUs) with an overall execution time of 18.6 ms for batch compared to 5.2 ms for MixUp on a A6000 Nvidia Quadro. This increase in execution training time is compensated, however, by an improvement in terms of accuracy (+3% and +5%, Table 1).
>
> **Q4: Dataset Diversity: Do you anticipate similar performance benefits when applying your augmentation strategy to significantly smaller or differently structured training datasets?**
>
> We conducted two additional experiments: (A) we used only 20% of our training set and (B) we trained on videos generated with a different synthetic generator (CogVideoX). Results for both experiments are presented below. In the first case we observe that our proposed augmentation strategy is still able to provide an advantage on a significantly smaller training set (+15% in terms of balanced accuracy) (second row).
>
> Table A: 20% of training set from Pyramid flow.
> | AUC / bAcc   |  Allegro    |  Cogvideo X |  Mochi-1    | OpenSora-Pl.|     AVG     |
> |:-------------|:-----------:|:-----------:|:-----------:|:-----------:|:-----------:|
> | without aug. | 94.3 / 70.8 | 89.2 / 62.8 | 93.9 / 77.8 | 98.4 / 87.3 | 94.0 / 74.7 |
> | with aug.    | 97.6 / 88.3 | 95.2 / 82.8 | 98.8 / 92.7 | 99.8 / 96.5 | 97.8 / 90.1 |
>
> In the second experiment we used CogvideoX, instead of Pyramid Flow, during training. In this case too, we observe a significant advantage of the proposed augmentation strategy (+5% in terms of AUC and +13% in terms of balanced accuracy) (second row).
>
> Table B: Using CogVideoX as the training set.
> | AUC / bAcc   |  Allegro    | Pyramid Flow|  Mochi-1    | OpenSora-Pl.|     AVG     |
> |:-------------|:-----------:|:-----------:|:-----------:|:-----------:|:-----------:|
> | without aug. | 90.7 / 69.3 | 98.0 / 87.3 | 90.7 / 72.0 | 94.9 / 78.0 | 93.6 / 76.7 |
> | with aug.    | 98.5 / 88.8 | 99.4 / 93.8 | 97.6 / 85.2 | 99.2 / 92.8 | 98.7 / 90.2 |

---

> > ### Comment · Reviewer_5ZLP · 2025-08-05
> >
> > I thank the authors for their thorough rebuttal and additional analysis, which have addressed most of my concerns. Accordingly, I maintain my rating as accept.

---

> > > ### Author Response · Authors · 2025-08-06
> > >
> > > Dear Reviewer 5ZLP,
> > >
> > > Thank you for your constructive comments.
> > >
> > > We are glad that our responses addressed your concerns and will incorporate your feedback in the final version of our paper.

---

### Official Review · Reviewer_BuLg · 2025-07-02

**Clarity:** 4
**Significance:** 3
**Originality:** 2
**Rating:** 5
**Confidence:** 4

**Summary:**

Recently, with the rapid development of video generative models, forensic detectors have also emerged; however, many of them lack generalizability. To address this issue, the authors propose a method that first identifies discriminative features that are less biased toward forensic-irrelevant patterns and more robust for detecting generated videos. Based on frequency domain analysis and previous literature, the authors identify that mid-high frequency components are less susceptible to compression artifacts and are thus suitable for forensic detection. To utilize these cues, two novel augmentation strategies are introduced during training: (1) Injection of forensic cues, by reconstructing real videos using the same autoencoder that generated the fake videos, aligning both real and fake samples in the representation space, and (2) Wavelet-based augmentation, by mixing real and fake videos using wavelet decomposition to enhance the mid-high frequency learning and push the detector to focus on those features. In the experimental evaluation, the model was trained using videos generated by the Pyramid Flow model, and testing was performed on both the GenVideo dataset (publicly available) and a newly constructed dataset comprising 2,400 videos from recent generative models. Accuracy was used as the evaluation metric, and the proposed method outperformed recent forensic detection methods.

**Questions:**

a. In lines 186–193, the authors state that replacing the low-frequency bands with those from the real counterpart forces the detector to focus on mid-high frequency features. However, how can we confidently claim that the detector will focus on mid-high frequencies? It is possible that the detector still learns from the low-frequency content provided by the real counterpart, potentially leading to misleading or incorrect detection. Do the authors have empirical evidence or observations supporting this assumption?
b. The proposed method appears to heavily rely on the reconstruction quality of the Pyramid Flow model. While the paper mentions that this model generally reconstructs videos without visible artifacts, what happens if it fails in certain cases? Would such failure compromise the robustness or generalizability of the detector trained using this reconstruction-based augmentation?
c. The paper makes valuable contributions, including a new dataset and a promising method. However, neither the code nor the dataset has been made available. The authors are strongly encouraged to release both the code and dataset to enable transparent evaluation, reproducibility, and faster progress in the rapidly evolving field of forensic video detection.

**Ethical Concerns:**

["NO or VERY MINOR ethics concerns only"]

**Final Justification:**

Thank you to the authors for the clarification on all the raised questions.

**Limitations:**

Yes

**Paper Formatting Concerns:**

No formatting issues.

**Quality:**

3

**Strengths And Weaknesses:**

1. Quality:
The submission is technically sound, presenting a relevant formulation and effective use of forensic feature extraction from videos. The claims are well-supported by experimental results. The methods employed are appropriate, and the work is presented as a complete and cohesive study. The authors are transparent about both the strengths and limitations of their approach.

2. Clarity:
The submission is clearly written and easy to follow.

3. Significance:
The results are impactful for the community. While the code and new dataset have not been released, if shared, this work has strong potential to be used by other researchers for further development and benchmarking.

4. Originality:
The work integrates several ideas from previous methods, with appropriate citations. Although the application of these ideas to video forensics is a novel context, the core techniques themselves are adapted rather than newly proposed. Therefore, the method is not entirely novel in its formulation.

---

> ### Author Rebuttal · Authors · 2025-07-30
>
> We thank the reviewer for their positive feedback on our paper. We are glad that they found our “submission technically sound”, the “claims well-supported by experimental results”, our “submission clearly written and easy to follow” and that our “results are impactful for the community”.
>
> In the following, we address each question individually.
>
> **Q1: The paper makes valuable contributions, including a new dataset and a promising method. However, neither the code nor the dataset has been made available. The authors are strongly encouraged to release both the code and dataset to enable transparent evaluation, reproducibility, and faster progress in the rapidly evolving field of forensic video detection.**
>
> We plan to publicly release both the dataset that we created with new synthetic generators and the code of our method to guarantee reproducibility, and to help advance the research field on video forensics.
>
> **Q2: a. In lines 186–193, the authors state that replacing the low-frequency bands with those from the real counterpart forces the detector to focus on mid-high frequency features. However, how can we confidently claim that the detector will focus on mid-high frequencies? It is possible that the detector still learns from the low-frequency content provided by the real counterpart, potentially leading to misleading or incorrect detection. Do the authors have empirical evidence or observations supporting this assumption?**
>
> To gain better insights into this issue we conducted the following additional experiment: at test time we modified the real videos and injected the mid-high frequencies of fully synthetic videos into them. The results of our detector with this test set are presented below. We observe that the results in terms of AUC and balanced accuracy (bAcc) get worse (second row). Moreover, in terms of the True Negative Rate (TNR), which is the probability that real videos are correctly classified as real, it goes from 98.7 to 0.2 (on average), which clearly shows that the detector inverts its decision. This highlights the importance of mid-high frequencies to establish if a video is fake or not.
>
> | AUC / bAcc |  Allegro    |  Cogvideo X |  Mochi-1    | OpenSora-Pl.|     AVG     |
> |:-----------|:-----------:|:-----------:|:-----------:|:-----------:|:-----------:|
> |            | 98.6 / 91.5 | 97.2 / 85.0 | 99.1 / 93.5 | 99.8 / 97.0 | 98.7 / 91.8 |
> | **         | 21.8 / 42.2 | 13.0 / 36.0 | 20.2 / 44.2 | 40.9 / 47.7 | 24.0 / 42.5 |
>
> **real videos with mid-high frequencies of fully-generated videos
>
> **Q3: The proposed method appears to heavily rely on the reconstruction quality of the Pyramid Flow model. While the paper mentions that this model generally reconstructs videos without visible artifacts, what happens if it fails in certain cases? Would such failure compromise the robustness or generalizability of the detector trained using this reconstruction-based augmentation?**
>
> Even though our method is trained on high-quality synthetic videos generated by the Pyramid Flow model (with a quality score above 84% on the VBench Leaderboard [30]), it is still able to detect videos with visible artifacts. This is evidenced by the fact that for videos generated by the ModelScope model that exhibit visible artifacts (with a quality score below 76% on the same benchmark [30]), our method achieves a high accuracy of 97.1%. Furthermore, overall, from our experiments we were not able to observe a different behaviour between older and newer synthetic generators (the former containing more visible artifacts) (see Table 3 and 4 of our paper). Our conjecture is that the detector is mostly focused on low level traces and is not very much influenced by semantic inconsistencies.

---

### Official Review · Reviewer_gFZ8 · 2025-07-03

**Clarity:** 4
**Significance:** 4
**Originality:** 3
**Rating:** 5
**Confidence:** 5

**Summary:**

This paper proposes a novel training paradigm for synthetic video detection using forensic-oriented data augmentation based on wavelet decomposition. The authors claim to identify discriminative features in mid-high diagonal frequencies that remain robust across different video generators against compression, and introduce wavelet-band replacement augmentation to redirect model focus from semantic artifacts to forensic cues. The method achieves 10+% bAcc improvement over existing detectors while only being trained on a single generator. The core contribution is to augment the data by preserving forensic artifacts while reducing semantic bias, enabling better cross-generator generalization.

**Questions:**

Minor suggestions:
- Missing detectors based on temporal forgery cues [1], motion decomposition [7], frequency analysis [5,6], augmentation [4], attention [8], and generalization [2,3]. Not all of them need to be compared against, but the related work section lacks comparison.
- Specifically for source generator detection for deepfake videos (line 137), there are more relevant work [7,9].
- A real challenge (and validation) would be evaluating the proposed method and others on Deepfake-eval-2024 dataset, which is claimed to be truly in the wild.
- Size of training/validation sets are significantly low.
- The order of the results is weird, talking about ablation before results is not conventional.
- Mixup and Cutmix cites can be moved to line 263.


[1] Guo, Z., Liu, Y., Zhang, J., Zheng, H., & Shan, S. (2025). Face Forgery Video Detection via Temporal Forgery Cue Unraveling. In Proceedings of the Computer Vision and Pattern Recognition Conference (pp. 7396-7405).

[2] Wang, Z., Bao, J., Zhou, W., Wang, W., & Li, H. (2023). Altfreezing for more general video face forgery detection. In Proceedings of the IEEE/CVF conference on computer vision and pattern recognition (pp. 4129-4138).

[3] Yan, Z., Zhao, Y., Chen, S., Guo, M., Fu, X., Yao, T., ... & Yuan, L. (2025). Generalizing deepfake video detection with plug-and-play: Video-level blending and spatiotemporal adapter tuning. In Proceedings of the Computer Vision and Pattern Recognition Conference (pp. 12615-12625).

[4] Yan, Z., Luo, Y., Lyu, S., Liu, Q., & Wu, B. (2024). Transcending forgery specificity with latent space augmentation for generalizable deepfake detection. In Proceedings of the IEEE/CVF Conference on Computer Vision and Pattern Recognition (pp. 8984-8994).

[5] Kashiani, H., Talemi, N. A., & Afghah, F. (2025). FreqDebias: Towards Generalizable Deepfake Detection via Consistency-Driven Frequency Debiasing. In Proceedings of the Computer Vision and Pattern Recognition Conference (pp. 8775-8785).

[6] Tan, C., Zhao, Y., Wei, S., Gu, G., Liu, P., & Wei, Y. (2024, March). Frequency-aware deepfake detection: Improving generalizability through frequency space domain learning. In Proceedings of the AAAI Conference on Artificial Intelligence (Vol. 38, No. 5, pp. 5052-5060).

[7] Demir, I., & Çiftçi, U. A. (2024). How do deepfakes move? motion magnification for deepfake source detection. In Proceedings of the IEEE/CVF Winter Conference on Applications of Computer Vision (pp. 4780-4790).

[8] Zhao, H., Zhou, W., Chen, D., Wei, T., Zhang, W., & Yu, N. (2021). Multi-attentional deepfake detection. In Proceedings of the IEEE/CVF conference on computer vision and pattern recognition (pp. 2185-2194).

[9] Çiftçi, U. A., Demir, I., & Yin, L. (2024). Deepfake source detection in a heart beat. The Visual Computer, 40(4), 2733-2750.

**Ethical Concerns:**

["NO or VERY MINOR ethics concerns only"]

**Final Justification:**

Almost all of my concerns are answered. I left a suggested additions list in my response. With those 7 items (or those that can fit in the main paper) it will be an interesting and meaningful contribution to NeurIPS.

**Limitations:**

Limitations of the approach are well discussed. The paper makes a valuable contribution to synthetic video detection by proposing a practical augmentation strategy that utilizes generation artifacts, claimed to be generalizable across multiple datasets and generators. The paper addresses an important problem with a technically sound approach and demonstrates promising results. With proper validation of compression robustness claims, technical details and ablation studies on the frequency bands, better comparison organization, and inclusion of recent baselines, this work could significantly advance the field's understanding of forensic-oriented detection methods.

**Quality:**

3

**Strengths And Weaknesses:**

Strengths:

- The foundation and motivation of the paper is very clear, connected to implementation and design choices, and related with the literature. The authors' focus on mid-high diagonal frequency components aligns with established research on GAN-specific frequency signatures. The choice of wavelet decomposition is well-justified, as wavelets preserve spatial information, while enabling multi-scale frequency analysis, while also being computationally more efficient.

- Forcing models to focus on intrinsic forensic cues rather than semantic differences is a strong motivation and is implemented clearly. It would have been even better if the paper documented (or referred) to the current landscape of detectors battling with this problem.

- Training only on one generator and generalizing to 15 generators is an excellent evaluation for generalization, especially for a production environment. The comparison tables, though, need another pass.

- Very interesting observations on the relation between compression and generative fingerprints, which leads to mid frequency components. A compression ablation study would be the cherry on the top.

Weaknesses:

- There needs to be a clear distinction between "generation-specific artifacts" vs. "artifacts introduced by specific generators", else the paper is trying to solve what is claimed to be solved by other methods.

- There is very little information on the denoiser, which can be very influential on the results. An ablation study with/without would enhance the design choice.

- To further emphasize the claims of the paper, I would be very interested to see a result where the same method is trained/tested on the training/testing set of all 15 generators. That would be a showstopper if training the same model on 15 generators performs worse than training on one augmented generator. If only Pyramid Flow training is causing such high performance, I would be excited to see the training on more generators. Then it opens the doors for yet another research question: is augmentation all we need (for df detection)? :)

- While it is known that compression helps with deepfake detection evasion as an adversarial strategy, the paper needs stronger empirical validation of compression robustness claims with systematic evaluation across multiple compression standards and quality levels, possibly with different frequency decompositions for the wavelet.

- I believe Tab. 3 is neither fair nor relevant for the claims of this paper. If the contribution of this paper is the new Wavelet-based augmentation method, then augmenting the training sets of these detectors and then observing their accuracy increase would be a fair comparison. If the contribution is the DINOv2 + augmented data, then it should have been trained/tested on the detectors' datasets. Tab. 4 comes closer to a fair ground, however the contribution is still unclear here: DINOv2 backbone itself is not making this difference, is it the augmented Pyramid Flow data + DINOv2? What happens if we augment the training data of all?

- I would assume compared detectors to have some commonality with this detector, may it be generalization, or fingerprints, or frequency-based analysis. There are many other detectors that might have been compared which share similar motivations, like other frequency domain approaches (DCT-based methods, Fourier analysis).

- While the paper identifies mid-high diagonal frequencies as discriminative, the analysis lacks depth compared to established frequency domain research. More in-depth analysis (or even classification) of spectral artifacts different generators or a quantitative frequency band analysis would ground the observations.

- The wavelet-band replacement augmentation procedure needs clearer algorithmic description.

---

> ### Author Rebuttal · Authors · 2025-07-30
>
> We thank the reviewer for their positive feedback on our paper. We are glad that they found that “it addresses an important problem”, its motivation is “very clear, connected to implementation”, and “related with the literature”, that our “approach is technically sound and results are promising”, our “observations on the relation between compression and generative fingerprints are very interesting” and that “training only on one generator and generalizing to 15 generators is an excellent evaluation”.
>
> In the following, we address each question individually.
>
> **Q1: "generation-specific artifacts" vs. "artifacts introduced by specific generators".**
>
> Thank you for pointing this out. We will make clearer the distinction between "generation-specific artifacts", which we consider to be traces related to the generation process, that are distinct from "artifacts introduced by specific generators", _e.g._, certain semantic visible cues, such as the lack of perspective or temporal consistency in the generated videos.
>
> **Q2: Denoiser used in visualization.**
>
> We used the denoiser only to extract the forensic fingerprints and to motivate our method. However, we do not use a denoiser in our detector, which is only based on the wavelet transform to implement our frequency-based augmentation strategy. We will make this clearer in the paper and give more details of the denoiser.
>
> **Q3: Training on 15 generators.**
>
> This is an interesting question. While including more generators might seem to improve performance, the answer is not straightforward. A key issue is which generators to include during training. For example, DeMamba [12] was trained on 10 generators (from 2023–2024) and performs well on older models but struggles to generalize to newer ones (see Table 3). This suggests that both the choice and the number of generators remain open research questions.
>
> It is also not feasible for us to apply our data augmentation procedure to many generators that require reconstruction with the same autoencoder used during generation. Nevertheless, as one datapoint, we conducted an experiment, wherein, we compared the model trained using two generators (Pyramid Flow (P) + CogVideoX (C)) with the same model trained using one single generator (P or C) and their corresponding versions with our wavelet-based augmentation. In the following table we show both AUC and balanced accuracy (bACC). We observe that using one single generator with its simpler design and our augmentation gives better results than including two different generators (92.5 and 95.3 vs 87.3 in terms of bACC). Arguably, adding more generators may potentially change these results. Nevertheless, it shows one quantified datapoint.
>
> | AUC / bAcc  |  Allegro  |  Mochi-1  |OpenSora-Pl|    Nova   |    Sora   |    Flux   |    AVG    |
> |:------------|:---------:|:---------:|:---------:|:---------:|:---------:|:---------:|:---------:|
> | C           | 90.7/69.3 | 90.7/72.0 | 94.9/78.0 | 99.1/94.2 | 89.0/63.5 | 94.6/77.5 | 93.2/75.8 |
> | C + our aug.| 98.5/88.8 | 97.6/85.2 | 99.2/92.8 | 99.8/95.8 | 99.8/96.8 | 99.7/95.3 | 99.1/92.5 |
> | P           | 94.7/68.2 | 94.9/75.5 | 98.0/80.0 | 99.0/87.7 | 95.7/69.5 | 96.8/72.0 | 96.5/75.5 |
> | P + our aug.| 98.6/91.5 | 99.1/93.5 | 99.8/97.0 | 99.3/93.2 | 99.9/98.5 | 99.9/98.0 | 99.4/95.3 |
> | P + C       | 94.9/80.5 | 96.5/86.5 | 98.7/94.7 | 99.4/96.5 | 94.4/77.5 | 96.8/88.2 | 96.8/87.3 |
>
> **Q4: Compression robustness.**
>
> We compressed the test dataset used in our ablation study (both real and fake videos) with H.264 at different quality levels by varying the Constant Rate Factor (CRF). Results using different wavelet decomposition levels are presented in the Table below and show that our method is robust to different compression levels with AUC always remaining above 80%.
>
> |     AUC / bAcc        |   2 levels  |   3 levels  |   4 levels  |
> |:----------------------|:-----------:|:-----------:|:-----------:|
> | CRF 20 (high quality) | 97.8 / 83.8 | 97.3 / 84.5 | 96.6 / 83.2 |
> | CRF 24                | 96.8 / 77.1 | 96.0 / 78.3 | 95.2 / 78.2 |
> | CRF 28                | 93.9 / 67.9 | 92.5 / 69.1 | 91.1 / 68.5 |
> | CRF 32 (low quality)  | 88.2 / 59.5 | 85.2 / 59.2 | 82.6 / 59.3 |
>
> The accuracy gets significantly lower when the quality decreases (around 60% in the last row, CRF 32) and this clearly indicates the need for a calibration procedure for low-quality compressed videos to estimate the optimal threshold value. By using only 10 compressed videos from the validation set for threshold calibration, we noted an increase in accuracy from 60% to 76%. We will include these results in the revised version of our paper.
>
> **Q5: Augmenting SoTA training data.**
>
> We understand this critique. However, it is not always obvious as to how to perform this analysis, since the training sets of current detectors can include generators whose architectures are not disclosed (e.g., Moonvalley, Pika), which means that we cannot apply our procedure that requires data reconstruction with the same autoencoder used during generation. In addition, some prior methods employ high-level semantic traces for detection. Hence our augmentation, which is focused on low-level traces makes less sense in their context.
>
> Nevertheless, we re-trained the method DMID [18] on our dataset and obtained an average improvement of +1.5 in AUC and +1.9 in Accuracy on the more recent generators (2024-25 years) by including our augmentation strategy. We will add more such comparisons with other prior datasets in the revised version of our paper.
>
> **Q6: Other frequency-based detectors.**
>
> There are indeed other methods that leverage frequency domain analysis to design an effective forensic detector. Some approaches aim at forcing the detector to focus on high-frequency information [A, B], while others exploit discrepancies in the whole frequency spectrum that can be achieved, _e.g._, by concatenating frequency-based components in the DCT domain [C]. A more recent work is based on the observation that the reconstruction error exhibits stronger mid-frequency components for real images versus for generated images [D]. In our approach, instead, we guide the detector to focus on diagonal mid-high frequencies, as video compression artifacts overlap with forensic cues in vertical/horizontal directions. Unlike [E], which uses frequency-domain Mixup without targeting model-specific cues, our method focuses on artifacts that persist after compression.
>
> We will add a paragraph in the Related Work section of the paper where we will explicitly make clear the differences between such approaches and our method. Furthermore, we will add comparisons with those works whose code is publicly available. We already performed some comparisons. Results are presented below in terms of AUC/bAcc both on their training data and by re-training on Pyramid Flow. Our method guarantees better performance than these detectors even when trained on the same dataset.
>
> | AUC / bAcc   |     Allegro |  CogvideoX  | OpenSora-Pl.|     Mochi-1 |        AVG  |
> |:-------------|:-----------:|:-----------:|:-----------:|:-----------:|:-----------:|
> | [A] FreqNet  | 75.7 / 66.7 | 42.9 / 45.5 | 81.0 / 68.3 | 54.4 / 52.8 | 63.5 / 58.3 |
> | [A] FreqNet* | 91.5 / 84.7 | 72.1 / 65.7 | 94.7 / 86.0 | 84.5 / 78.0 | 85.7 / 78.6 |
> | [D] Fire     | 60.5 / 51.3 | 53.8 / 49.8 | 88.5 / 51.5 | 81.8 / 51.3 | 71.1 / 51.0 |
> | [D] Fire*    | 84.0 / 75.5 | 68.6 / 66.0 | 66.6 / 63.5 | 79.7 / 73.0 | 74.8 / 69.5 |
> |     Ours     | 98.6 / 91.5 | 97.2 / 85.0 | 99.8 / 97.0 | 99.1 / 93.5 | 98.7 / 91.8 |
> (*) re-trained on Pyramid Flow
>
> References:
> [A] Tan et al. Frequency-Aware Deepfake Detection: Improving Generalizability through Frequency Space Domain Learning, AAAI 2024
> [B] Wang et al. Dynamic Graph Learning with Content-guided Spatial-Frequency Relation Reasoning for Deepfake Detection, CVPR 2023
> [C] Qian et al. Thinking in Frequency: Face Forgery Detection by Mining Frequency-aware Clues, ECCV 2020
> [D] Chu et al. FIRE: Robust Detection of Diffusion-Generated Images via Frequency-Guided Reconstruction Error, CVPR
> 2025
> [E] Kashiani et al. FreqDebias: Towards Generalizable Deepfake Detection via Consistency-Driven Frequency Debiasing, CVPR 2025
>
> **Q7: More in-depth analysis.**
>
> To support the conjecture that the mid-high diagonal frequencies are more discriminative, we conducted a further analysis in Section B of the Appendix, where we compare real and reconstructed videos in the frequency domain by computing the distance defined in equation 7 (see supplemental material). A higher value for this distance indicates a greater difference between the real videos and the reconstructed ones, and therefore a more discriminative capability of the corresponding frequency. For mid-high diagonal frequencies the values concentrate around 89%, while for horizontal and vertical frequencies it is equal to 11%, indicating that the former frequencies are more discriminative.
>
> To further compare to the established frequency domain research, we extracted mid-frequency and high-frequency content, which is mostly used in related work for discrimination [A,D]. In such regions the distance values concentrate around 55% and 68%, respectively. As a conclusion, the difference between real and fake videos is more pronounced at mid-high diagonal frequencies.
>
> **Q8: Writing and organization.**
>
> We will improve the algorithmic description of our method; add all suggested papers in the Related Work Section and change the order of the results.
>
> **Q9: Deepfake-eval-2024 dataset.**
>
> We ran our detector on the fully synthetic videos, obtaining an AUC of 82.4%.
>
> **Q10: Size of training/validation sets are significantly low.**
>
> This is true, however we do not see this as a weakness since our augmentation helps to compensate for that (this is also confirmed by the experiment we conducted in response to Q3).

---

> > ### Author Response · Authors · 2025-08-06
> >
> > Dear Reviewer gFZ8,
> >
> > Thank you for reviewing our paper and for the constructive comments.
> >
> > We would like to follow up to ensure that our responses have adequately addressed the main concerns you raised.
> > If further clarification is needed, we would be more than happy to elaborate.

---

### Decision · Program_Chairs · 2025-09-17

**Decision:**

Accept (poster)

**Comment:**

This manuscript addresses the task of generalizable AI-generated video detection. It proposes a forensics-oriented data augmentation strategy which is based on a wavelet decomposition, and augmenting particular frequency bands such as to encourage detectors to learn to recognize forensic traces that are consistent across various generative architectures and robust to video compression artifacts.

Reviewers highlight the clear motivation of the approach and the promising results. All reviews agree that the paper should be accepted. The AC agrees, yet strongly recommends to discuss related work that reduces frequency artifacts in the generator right away as well as previous works discussing generalizable generated image detection beyond artifacts, for example:

Durall et al: Watch your up-convolution: Cnn based generative deep neural networks are failing to reproduce spectral distributions, CVPR 2020
He et al.: Beyond the Spectrum: Detecting Deepfakes via Re-Synthesis, IJCAI 2021
Jung et al.: Spectral Distribution aware Image Generation, AAAI 2021